# DHX9-dependent recruitment of BRCA1 to RNA promotes DNA end resection in homologous recombination

Prasun Chakraborty [1] & Kevin Hiom [1]✉

Double stranded DNA Breaks (DSB) that occur in highly transcribed regions of the genome are preferentially repaired by homologous recombination repair (HR). However, the mechanisms that link transcription with HR are unknown. Here we identify a critical role for DHX9, a RNA helicase involved in the processing of pre-mRNA during transcription, in the initiation of HR. Cells that are deficient in DHX9 are impaired in the recruitment of RPA and RAD51 to sites of DNA damage and fail to repair DSB by HR. Consequently, these cells are hypersensitive to treatment with agents such as camptothecin and Olaparib that block transcription and generate DSB that specifically require HR for their repair. We show that DHX9 plays a critical role in HR by promoting the recruitment of BRCA1 to RNA as part of the RNA Polymerase II transcription complex, where it facilitates the resection of DSB. Moreover, defects in DHX9 also lead to impaired ATR-mediated damage signalling and an inability to restart DNA replication at camptothecin-induced DSB. Together, our data reveal a previously unknown role for DHX9 in the DNA Damage Response that provides a critical link between RNA, RNA Pol II and the repair of DNA damage by homologous recombination.

[1] Division of Cellular Medicine, Jacqui Wood Cancer Centre, School of Medicine, University of Dundee, Scotland, United Kingdom. ✉email: k.hiom@dundee.ac.uk

DNA breaks pose a significant threat to genomic stability and if repaired improperly can lead to increased mutagenesis, the generation of harmful chromosome aberrations, and cell death. Defects in the repair of DSB have been linked with a wide variety of diseases including immunodeficiency, neurodegenerative disease, hematological disease, progeria, and cancer.

DSB are generally repaired by one of two pathways: non-homologous end-joining, which is error prone and homologous recombinational repair that is relatively error-free. Although many of the biochemical steps required in the physical repair of DSB have been well characterized, the factors that govern the regulation of these pathways and the genomic context in which they operate is less well understood. For example, it was recently demonstrated that while DSB that occur in highly transcribed regions of the genome are preferentially repaired through HR[1], those in transcriptionally inactive loci are not. However, the mechanism through which HR is targeted to transcriptionally active regions of the genome remains unknown.

Interestingly, in addition to the well-characterized range of DNA binding proteins that are recruited at or near sites of DNA damage, a variety of RNA binding proteins (RBP) also accumulate at DSB where they are hypothesized to facilitate DNA repair[2]. The recruitment of RBPs is often linked to the generation of a variety of long and short RNA transcripts in the vicinity of DNA damage as part of the DNA damage response (DDR)[3]. While several hypotheses have been proposed to explain how RNA and RBP might contribute to the repair of DNA lesions at actively transcribed genes, a consensus mechanism has yet to emerge.

DHX9 is a multifunctional DNA/RNA helicase and a component of the RNA polymerase II (RNA Pol II) holoenzyme where it contributes to co-transcriptional processing of pre-mRNA[4]. Efforts to knock out the DHX9 gene in various cell lines using Crispr/Cas9 have failed and homozygous deletion of DHX9 in mice results in embryonic lethality[5], indicating that its function is essential. However, cells depleted of DHX9 using siRNAs exhibit relatively normal viability, suggesting that a residual low level of DHX9 is sufficient to sustain cell proliferation[6].

In addition to its roles in RNA splicing and RNA editing, several observations point to a role for DHX9 in DNA replication and/or the maintenance of genome stability. For instance, in cells DHX9 localizes to origins of replication and its suppression in human diploid fibroblasts leads to blocked replication, P53-mediated arrested growth, and senescence[7]. It also interacts with components of the DDR such as BRCA1[8] and Ku86[9] that are important for the repair of DNA breaks caused by replication stress.

DHX9 is an SF2 type helicase with functional similarities to the DExH-box DNA repair helicases BLM and WRN[10]. Biochemical studies established that DHX9 binds to single-stranded DNA and RNA and, like BLM and WRN, unwinds abnormal nucleic acid secondary structures that form during DNA replication and transcription[6]. These substrates include RNA-DNA hybrid, D-loops, and DNA forks, as well as RNA and DNA guanine quadruplexes (G4-DNA/RNA)[11,12], all of which have the potential to cause genomic instability. Most recently DHX9 was shown to be required for resolving large secondary structures that occur in transcripts of inverted genomic Alu repeats[13]. Nevertheless, a specific role for DHX9 in the repair of DNA damage has yet to be established.

Here we demonstrate that DHX9 is a bone fide component of the DDR and show how it plays a critical role in linking transcription to the repair of DNA breaks by HR.

## Results

**DHX9 is redistributed in response to DNA damage.** Proteins involved in the DDR are often recruited at or near sites of DNA damage to form nuclear foci. To establish if DHX9 is a bone fide

DDR protein we investigated whether it also accumulates in nuclear foci in cells with damaged DNA. To do this we treated cells with two agents, camptothecin (Cpt) and ionizing radiation (IR), that introduce DSB into the genome through different mechanisms (Supplementary Fig. 1). IR introduces DSB into the genome through the collision of high-energy particles that directly break the sugar-phosphate backbone of DNA. It also splits water molecules to create hydroxyl radicals that similarly attack the DNA backbone to generate both DSB and single-stranded DNA breaks (SSB). Cpt, on the other hand, inhibits topoisomerase I (top I) leading to the generation of SSB in which the 3′ end of the break is covalently attached to top I protein. This is referred to as a top I cleaved complex (top Icc)[14]. Replication forks encountering these lesions are prone to stalling and collapse, converting SSB to DSB. Since top I plays a pivotal role in the removal of supercoils during transcription, Cpt-induced DSB are commonly enriched in transcribed regions of the genome.

We treated cells with IR and Cpt and then stained them with a specific antibody, using fluorescence imaging to determine the distribution of DHX9 protein in the nucleus. Importantly, we first confirmed that our antibody was specific for DHX9 by showing that it detected the protein in cells transfected with a nonspecific control siRNA but not in cells in which DHX9 was knocked down using a specific siRNA (Supplementary Fig. 2a).

In cells treated with Cpt, DHX9 accumulated in discreet nuclear aggregates/foci that were not observed in undamaged cells (Fig. 1a). Importantly, DHX9 foci colocalized on chromatin with the phosphorylated form of histone H2AX (γH2AX), confirming that it is recruited at or near sites of DNA breaks (Fig. 1b). While DHX9 aggregates were also detected in cells exposed to IR, these foci did not significantly co-localize with γH2AX (Fig. 1b), indicating that DHX9 is not recruited equally at all DSB.

Although DHX9 accumulated at DSB in cells that were treated with Cpt for 30 min, these foci were relatively faint (Supplementary Fig. 1b). However, DHX9 foci stained more intensely after 2 to 4 h Cpt treatment, suggesting that molecules of DHX9 accumulate at DSB over time. Since DHX9 associates with the transcription complex we hypothesized that the increased intensity in DHX9 staining might reflect the accumulation of stalled RNA polymerase II (RNA Pol II) at DSB. Consistent with this hypothesis, we found that Cpt-induced DHX9 foci were greatly diminished in cells that were also treated with the transcription inhibitor DRB (5,6-dichloro-1-β-D-ribofuranosyl-benzimidazole) (Fig. 1c). Furthermore, DHX9 foci induced by both Cpt and IR were disassembled when cells were treated with RNaseA to degrade RNA (Fig. 1d and Supplementary Fig. 2c, d). Taken together, these data indicate that DHX9 accumulates at Cpt-induced DSB through a mechanism that is dependent on RNA and RNA Pol II transcription. We speculate that the accumulation of DHX9 in IR-induced foci also reflects stalling of RNA Pol II, but at lesions other than DSB.

DHX9 has been linked to the metabolism of R-loops that form during transcription at sites of paused RNA Pol II[6,15]. Stalling of RNA Pol II enables the nascent RNA to invade its complementary duplex DNA, generating a region of DNA-RNA hybrid and a displaced ssDNA. DNA-RNA hybrid has been hypothesized to play a role in the repair of DSB through an, as yet, unspecified mechanism. To determine whether the generation of R-loops was linked to the formation of DHX9 foci we treated cells with Cpt and prior to imaging incubated them with RNAseH1 to specifically degrade DNA-RNA hybrid (Supplementary Fig. 2b, e). The faint DHX9 foci that formed after 30 min exposure to Cpt were resistant to treatment with RNaseH1, suggesting that these early forming foci were not associated with R-loops (Supplementary Fig. 2b). Intriguingly, the intense DHX9 foci that formed after two or more hours of exposure to Cpt were greatly

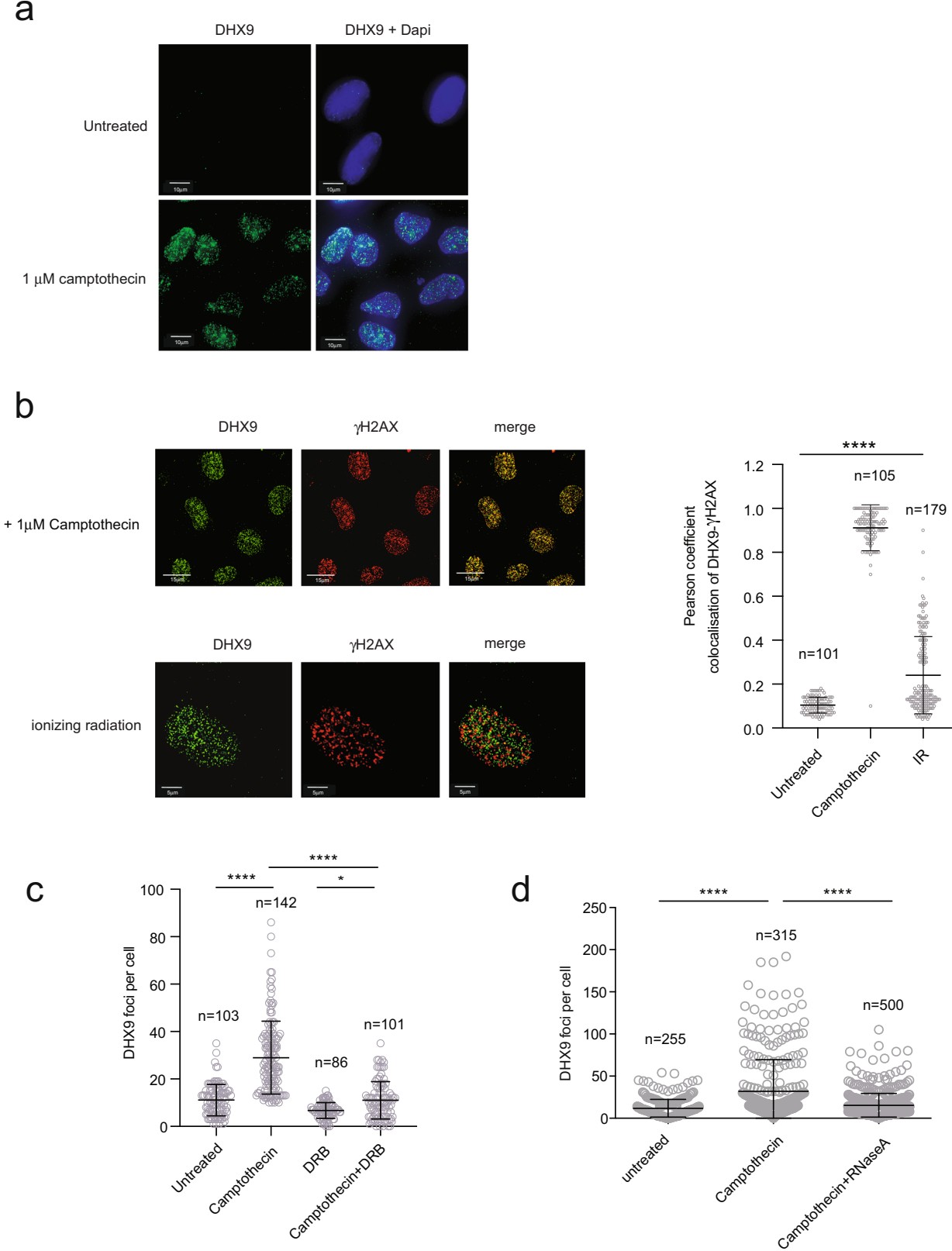

diminished by RNaseH1 treatment, indicating that DNA-RNA hybrid was present at these sites (Supplementary Fig. 2e). This supported our hypothesis that Cpt-induced DHX9 foci develop over time and that the more intense, later forming DHX9 foci represent DSB at which stalling of transcription complexes is associated with the generation of DNA-RNA hybrid. Whether R-

loops contribute to the repair of DSB remains the subject of much speculation.

To establish whether DHX9 is required for the repair of DSB we used two different siRNAs to deplete DHX9 protein and measured the sensitivity of these cells to different DNA damaging agents (Fig. 2a). Importantly, we first confirmed that knockdown

**Fig. 1 DHX9 accumulates at sites of DNA damage. a** Fluorescence image showing that DHX9 localizes to DNA damage-induced nuclear foci in cells treated with 1 μM camptothecin for 4 h. Nuclei are stained with Dapi. **b** Fluorescence images showing that DHX9 (green) co-localizes with γH2AX (red) in cells treated with 1 μM camptothecin for 4 h but not in cells treated with 10 Gy Ionizing radiation (left panel). Merged images identify co-localized proteins as staining yellow. Quantification of colocalization of DHX9 and γH2AX treated with 1 μM camptothecin and 10 Gy ionizing radiation (right panel) plotted as Pearson coefficient. **c** Localization of camptothecin-induced DHX9 in foci is impaired in cells treated with 10 μM DRB to inhibit transcription. **d** DHX9 damage-induced foci disintegrate after treatment with RNaseA. Quantification of $n$ cells (as indicated) from three pooled biologically independent experiments were performed in (**b–d**). Statistical significance was determined using one-way ANOVA with Tukey's post hoc test (****$p < 0.0001$, *$p < 0.1$). Mean and error bars indicating one standard deviation are also indicated. Source data are provided as a Source Data file.

of DHX9 did not cause alterations to the cell cycle that might influence the ability of these cells to repair DNA damage (Supplementary Fig. 3a). We then showed that depletion of DHX9 in cells caused hypersensitivity to both Cpt and Olaparib, an inhibitor of poly-ADP ribose polymerase 1 (PARP1) (Fig. 2b, c). However, depletion of DHX9 did not sensitize cells to IR-induced DNA damage (Fig. 2d). This directly mirrored our experiments showing that DHX9 accumulated at DSB induced by Cpt but less at IR-induced DSB (above). Importantly, the hypersensitivity of DHX9 deficient cells to Cpt and to Olaparib was reversed by expression of wild-type DHX9 from a plasmid but not the helicase "defective" DHX9 D511A E512A mutant protein (DHX9dead)[16], indicating that the helicase activity of DHX9 is required in the repair of DNA damage (Fig. 2b, c).

**DHX9 is required for homology dependent repair of DNA breaks.** DSB are generally repaired by two major pathways; non-homologous end joining (NHEJ) and HR. To establish whether DHX9 contributes to the repair of DSB by these pathways we used two established GFP-based reporter assays (cell lines H1299-dA3-1[17] and U2OS pDR-GFP) that measure the repair of DSB induced by expression of the restriction endonuclease I-Sce1 (Fig. 3a, b). As expected, siRNA-mediated depletion of the end-joining proteins Ku86 and p54nrb in H1299-dA3-1 cells caused a reduction in the repair of I-Sce1 induced DSB by NHEJ, compared to a siRNA control (Fig. 3a). Interestingly, H1299-dA3-1 cells depleted for DHX9 were not only proficient in NHEJ but exhibited elevated repair of I-Sce1 induced breaks through this pathway. We hypothesized that the depletion of DHX9 stimulated NHEJ by inhibiting the repair of DSB through a competing pathway, most likely HR. Importantly, we confirmed that elevated NHEJ was specific to the loss of DHX9 in these cells, by showing that overexpression of a siRNA resistant myc-DHX9 gene from a plasmid reduced NHEJ to control levels.

On the other hand, knockdown of DHX9 in U2OS cells containing the pDR-GFP HR reporter led to a 50–70% reduction in the repair of DSB by HR, compared to wild-type cells (Fig. 3b). This HR defect was comparable to that of cells deficient for the key HR mediator protein BRCA1. Critically, HR was restored to cells that overexpressed a siRNA-resistant myc-DHX9 cDNA from a plasmid (Fig. 3c). These data confirmed that DHX9 is required for the repair of DSB by HR and to prevent the hypersensitivity of cells to agents such as Cpt and Olaparib.

**DHX9 promotes DNA end resection.** Repair of DSB by HR requires that broken DNA ends are first resected to generate regions of ssDNA. Single-stranded DNA ends are protected from degradation by the binding of the single-strand binding protein complex RPA, which is subsequently replaced by the RAD51 recombinase to catalyze DNA strand exchange and drive HR. To determine whether DHX9 is required in these early steps of HR we treated cells with IR and with Cpt and used fluorescence imaging to detect γH2AX and RPA as measures of DNA breaks and DNA end-resection, respectively. Cpt treatment induced

similar levels of γH2AX foci in wild-type cells and in cells depleted of DHX9, reflecting the generation of equivalent numbers of DSB (Fig. 4a and Supplementary Fig. 4a, b). However, cells that were depleted of DHX9 had many fewer RPA foci than did wild-type cells, suggesting that DHX9 is required in DNA resection (Fig. 4b and Supplementary Fig. 3b). We confirmed that resection was dependent on the helicase activity of DHX9 DNA by showing that RPA foci were restored to cells in which a wild-type siRNA-resistant DHX9 cDNA was overexpressed from a plasmid, but not in cells expressing cDNA encoding the helicase defective DHX9 D511A/E512A mutant protein (Fig. 4c). Western blot analysis also confirmed that chromatin-bound RPA was greatly reduced in cells depleted of DHX9 compared to control cells (Fig. 4d and Supplementary Fig. 3c). Finally, we used an established BrdU staining assay to quantify ssDNA by fluorescence imaging[16]. This confirmed that the generation of ssDNA was significantly impaired in DHX9 defective cells compared to wild type (Supplementary Fig. 3d). Resection of DNA ends and binding of RPA are prerequisite for the recruitment of RAD51 to sites of DNA damage. Accordingly, DHX9 deficient cells were also greatly impaired in the formation of RAD51 foci in response to Cpt-induced DNA damage and exhibited reduced binding of RAD51 to chromatin (Fig. 5a and Supplementary Fig. 3c). These data, together with our data showing that DHX9 localizes to sites of DNA damage, established that recruitment of DHX9 at or near to DSB helps to suppress NHEJ and promote DNA resection.

Although cells depleted of DHX9 were also impaired in the recruitment of RPA and RAD51 to IR-induced DNA breaks (Supplementary Fig. 4c–f), this defect was markedly reduced compared with cells treated with Cpt. This was consistent with our earlier finding that colocalization of DHX9 with γH2AX is significantly less in cells damaged with IR. While it is possible that the pathways for resecting DNA breaks generated by Cpt and IR might differ in their requirement for DHX9, the difference might also reflect a greater contribution for NHEJ in the repair of IR-induced DSB compared with Cpt induced breaks.

Classical resection of DNA ends is initiated by CTIP[18,19] and the MRN (MRE11-RAD50-NBS1) nuclease complex and then extended in the 5′-3′ direction by EXO1 exonuclease or through a combination of DNA2 and BLM (reviewed in ref. [20]). Therefore we next determined whether DHX9 was in the same genetic pathway for DNA resection as CTIP and MRE11. Cpt-induced recruitment of RPA to foci was greatly reduced in cells knocked down for CTIP, confirming that the majority of DNA resection is dependent on the activity of CTIP (Fig. 5b). This was broadly comparable with the resection defect observed in cells knocked down for DHX9 (Fig. 5b and Supplementary Fig. 3d). Since the mean number of RPA foci was not further diminished in cells depleted of both CTIP and DHX9, we concluded that DHX9 and CTIP probably function in a common pathway for the generation of ssDNA. It was previously reported that in cells that are defective in RNA splicing, impaired DNA end resection may result from the decreased expression of CTIP[21]. Importantly, we confirmed that knockdown of DHX9 did not affect cellular levels

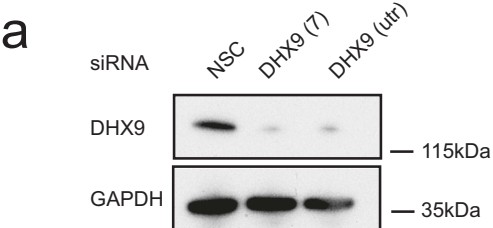

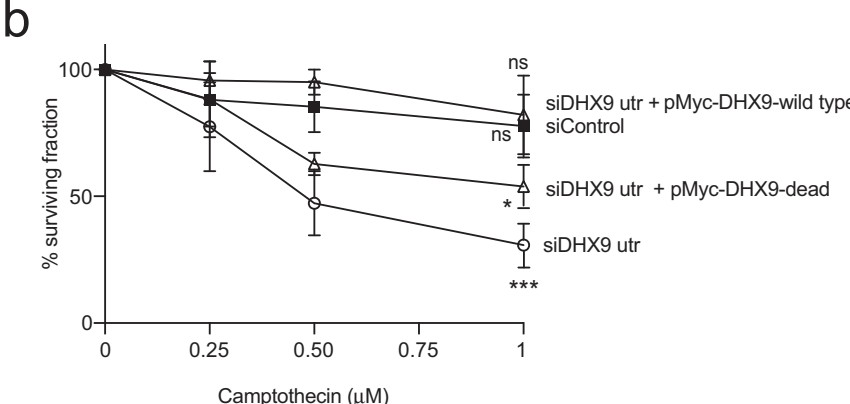

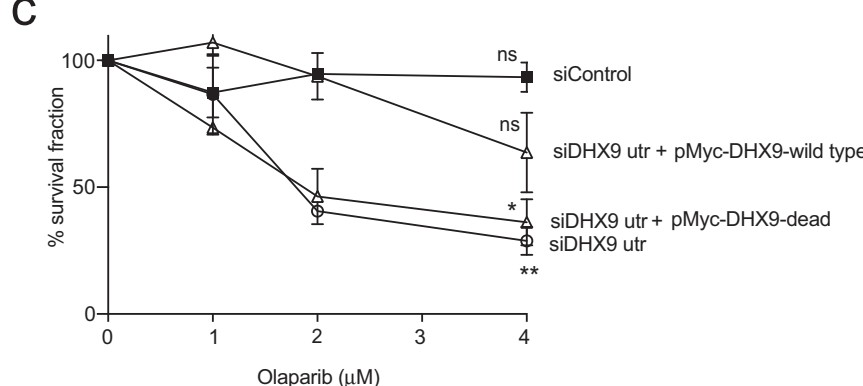

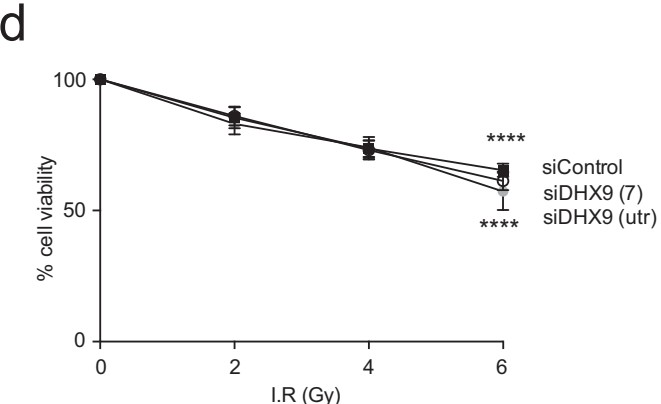

**Fig. 2 DHX9 is required for the repair of DNA damage induced by camptothecin and olaparib. a** Western blot showing knockdown of DHX9 by two different siRNAs, DHX9(7) and DHX9(utr) in U2OS cells. GAPDH is indicated as a loading control. **b** Clonogenic survival assay showing that cells knocked down for DHX9 are hypersensitive to DNA damage caused by treating cells with camptothecin and **c** Olaparib. **d** Graph of percentage cell viability after treatment of indicated cells with ionizing radiation measured by counting live cells in a Casey Cell counter. Cells depleted of DHX9 exhibit no decrease in viability compared to wild-type cells. Graphs include data from three biologically independent experiments. Statistical significance was determined using one-way ANOVA with Tukey's post hoc test (****$p < 0.0001$, ****$p < 0.0001$, ***$p < 0.001$, *$p < 0.1$, ns not significant). Mean and error bars indicating one standard deviation are also indicated. Source data are provided as a Source Data file.

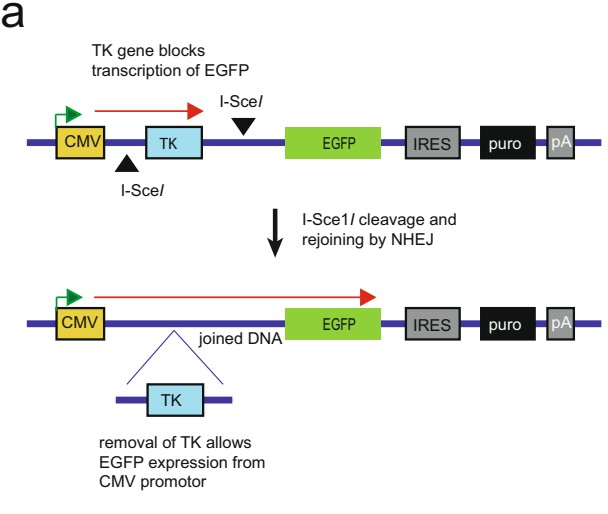

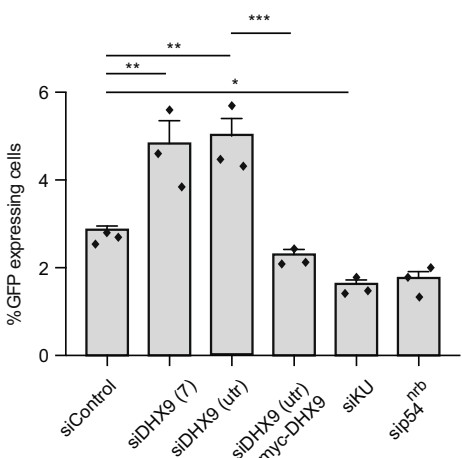

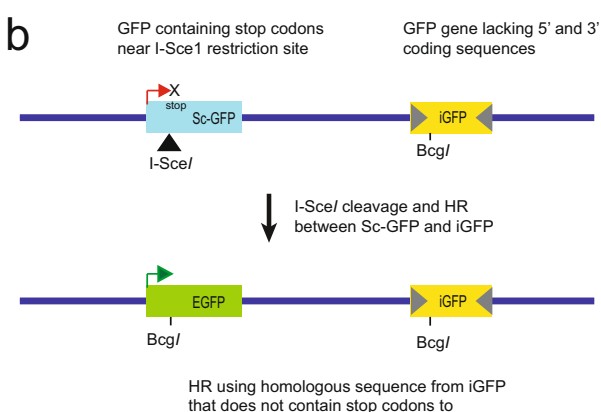

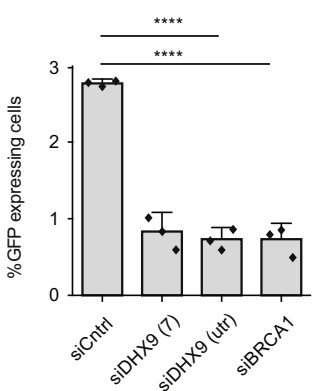

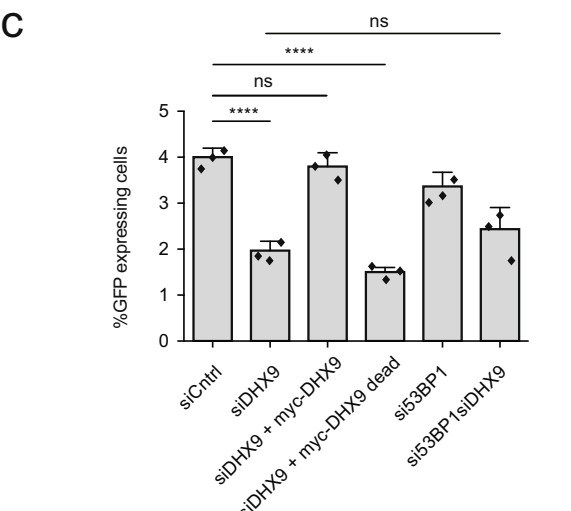

of CTIP, RAD51, or BRCA1 in our experiments (Supplementary Fig. 5a).

We then confirmed that MRE11 was also required for the generation of RPA in response to Cpt-induced DNA damage. Interestingly, the contribution of MRE11 to the formation of RPA foci was less than that of either DHX9 or CTIP (Fig. 5b).

Nevertheless, because DNA resection was similar in cells depleted of both MRE11 and DHX9 to cells impaired in DHX9 alone, we concluded that MRE11 and DHX9 also function in a common genetic pathway for DNA resection. Together, these data suggest that DHX9 is required for classical end-resection of broken DNA ends with both CTIP and MRE11 that together facilitate the

**Fig. 3 DHX9 is required in DSB repair by homologous recombination. a** DHX9 deficient cells are proficient in the repair of I-*Sce1* induced DSB by non-homologous end joining (NHEJ). Left panel: NHEJ reporter system in which NHEJ mediated repair of two DSB leads to the deletion of the thymidine kinase gene enabling expression of GFP (red arrow) from the CMV promotor (green arrow). Right panel: Graph of percentage of cells expressing GFP as a measure of successful NHEJ. Knockdown of key NHEJ proteins Ku86 and p54nrb in reporter cell line H1299-dA3-1 was performed as a positive control. Elevated NHEJ in DHX9 deficient cells is suppressed by the expression of siRNA-resistant myc-DHX9 expressed from a plasmid. **b** U2OS cells knocked down for DHX9 are deficient in the repair of I-*Sce1* induced DSB by recombination (HR) using a pDR-GFP U2OS reporter assay. Left panel: Reporter measuring HR mediated repair of an I-*Sce1* induced DSB in a GFP gene (Sc-GFP) containing multiple stop codons, using homologous sequences in a defective iGFP gene with 3′ and 5′ terminal deletions. Only repair of the DSB by HR generates an active EGFP gene. Right panel: Cells knocked down for DHX9 and BRCA1 with different siRNAs. Percent GFP-expressing cells are shown. **c** HR-mediated DSB repair requires the helicase activity of DHX9. Overexpression of siRNA resistant myc-tagged wild-type DHX9, but not helicase "dead" mutant DHX9 restores HR mediated repair to cells knocked down with siRNA DHX9(utr) targeted against the non-translated region of DHX9. Knockdown of 53BP1 in DHX9 defective cells does not restore HR. In (**a**), (**b**), and (**c**) mean values from $n = 3$ independent experiments are shown to be statistically significant using one-way ANOVA test with Tukey's post hoc analysis (ns not significant, $*p < 0.1$, $**p < 0.01$, $***p < 0.001$, and $****p < 0.0001$). Error bars of 1 SD are also indicated. Source data are provided as a Source Data file.

recruitment of downstream proteins such as RAD51 that are essential for HR mediated DSB repair.

**DHX9 responds to signaling by ATR and ATM.** To establish whether the recruitment of DHX9 in response to Cpt-induced DNA damage was dependent on DNA damage signaling by the ATM and ATR kinases, we measured recruitment of DHX9 to DNA damage in the presence and absence of the ATR and ATM inhibitors VE-821 and KU55933. This revealed that while DHX9 foci were diminished in cells treated with either ATR or ATM inhibitors, the effect of inhibiting ATR was greater than inhibiting ATM (Fig. 5c).

We also looked at the effect of these inhibitors on the accumulation of RPA as a marker of DNA end resection. In wild-type cells, inhibition of ATR reduced the formation of RPA foci to a low level, similar to that observed in cells knocked down for DHX9 (Fig. 5d). Since knockdown of DHX9 and inhibition of ATR together caused no further reduction in RPA foci we inferred that DHX9-mediated end resection responded to the ATR signaling pathway. A similar relationship was observed in cells where ATR was knocked down using siRNA instead of with chemical inhibition (Fig. 5e).

Although knockdown of ATM in U2OS cells also impaired the generation of RPA foci in response to Cpt-induced DNA damage (Fig. 5e), this defect was less severe than that caused by depletion of either DHX9 or ATR. This suggested that ATR may play a bigger role than ATM to promote DNA end resection in response to Cpt-induced DNA damage. Nevertheless, knockdown of DHX9 was epistatic to ATM for the recruitment of RPA, suggesting that these proteins also function in a common pathway.

Previous studies have shown that DNA damage signaling by ATR requires its autophosphorylation on Threonine 1989[22]. To determine if DHX9 was required for ATR activation we measured DNA damage-induced autophosphorylation in cells that were knocked down for DHX9. Whereas, in wild-type cells, Cpt treatment induced robust phosphorylation of ATR on Threonine 1989, this was greatly diminished in cells depleted for DHX9 (Fig. 5f). Moreover, phosphorylation of Chk1, a downstream target for ATR, was also diminished. Hence, DHX9 and ATR work together in a common pathway for the resection of DSB during HR.

**DHX9 interacts with BRCA1 in response to DNA damage.** The repair of DNA breaks by HR is dependent on the mediator protein BRCA1[23]. Cells that are defective in BRCA1, like those depleted of DHX9, are impaired in DNA resection and fail to recruit RAD51 to sites of DNA damage[24,25]. To establish if

BRCA1 and DHX9 work together in HR we depleted both proteins in U2OS pDR-GFP cells and measured HR mediated DSB repair. Since the HR defect in cells lacking both BRCA1 and DHX9 was similar to that of cells knocked down for DHX9 and BRCA1 individually, we concluded that these two proteins operate in a common genetic pathway for HR (Fig. 6a). Consistent with this, we also observed significant co-localization of BRCA1 with DHX9 in DNA-damage-induced foci in cells treated with Cpt (Fig. 6b).

It was previously reported that recombinant DHX9 binds to a c-terminal fragment of BRCA1 in GST pull-down experiments and in a yeast two-hybrid assay[8,26]. To determine whether endogenous BRCA1 and DHX9 interact in cells, we immuno-precipitated DHX9 and looked for co-purification of BRCA1. Surprisingly, very little endogenous BRCA1 co-immunoprecipitated with DHX9 in unperturbed cells, suggesting that there might be limited interaction between DHX9 and BRCA1 in the absence of DNA damage, or that the interaction is unstable (Fig. 7a). In contrast, DHX9 and BRCA1 interacted robustly in cells that were treated with DNA damaging agents (Cpt, hydroxyurea, and IR). Importantly, co-purification of BRCA1 with DHX9 was not mediated through RNA as all immunoprecipitations were performed in the presence of RNaseA. This indicated that BRCA1 and DHX9 probably interact prior to their localization at DNA damage. We previously found that the detection of DHX9 on chromatin was enhanced at promoter regions and at transcription termination sites at which RNA Pol II was stalled[6]. Therefore we speculate that DHX9 and BRCA1 might interact constitutively, but that detection of this complex by immunoprecipitation is enhanced when RNA Pol II is stalled at sites of DNA damage.

DHX9 is thought to bridge the interaction between BRCA1 and RNA Polymerase II holoenzyme[8]. We recently showed that DHX9 promotes the recruitment of various proteins to RNA Pol II holoenzyme by facilitating their binding to nascent RNA[6]. Therefore, we next addressed whether the interaction of BRCA1 and DHX9 with the RNA Pol II holoenzyme was dependent on RNA. To do this we immunoprecipitated RNA Pol II and measured the co-purification of DHX9 and BRCA1 in the presence and absence of RNaseA. While DHX9 and BRCA1 co-immunoprecipitated with RNA Pol II, BRCA1 was largely absent in samples that were treated with RNaseA, with only a very small amount of DHX9 retained (Fig. 7b). This confirmed that the association of both BRCA1 and DHX9 with RNA Pol II is mediated primarily through RNA, most likely the nascent strand formed during RNA Pol II-mediated transcription. Interestingly, co-purification of BRCA1 and DHX9 with RNA Pol II was significantly diminished in cells in which either protein was depleted. This indicated that BRCA1 and DHX9 probably

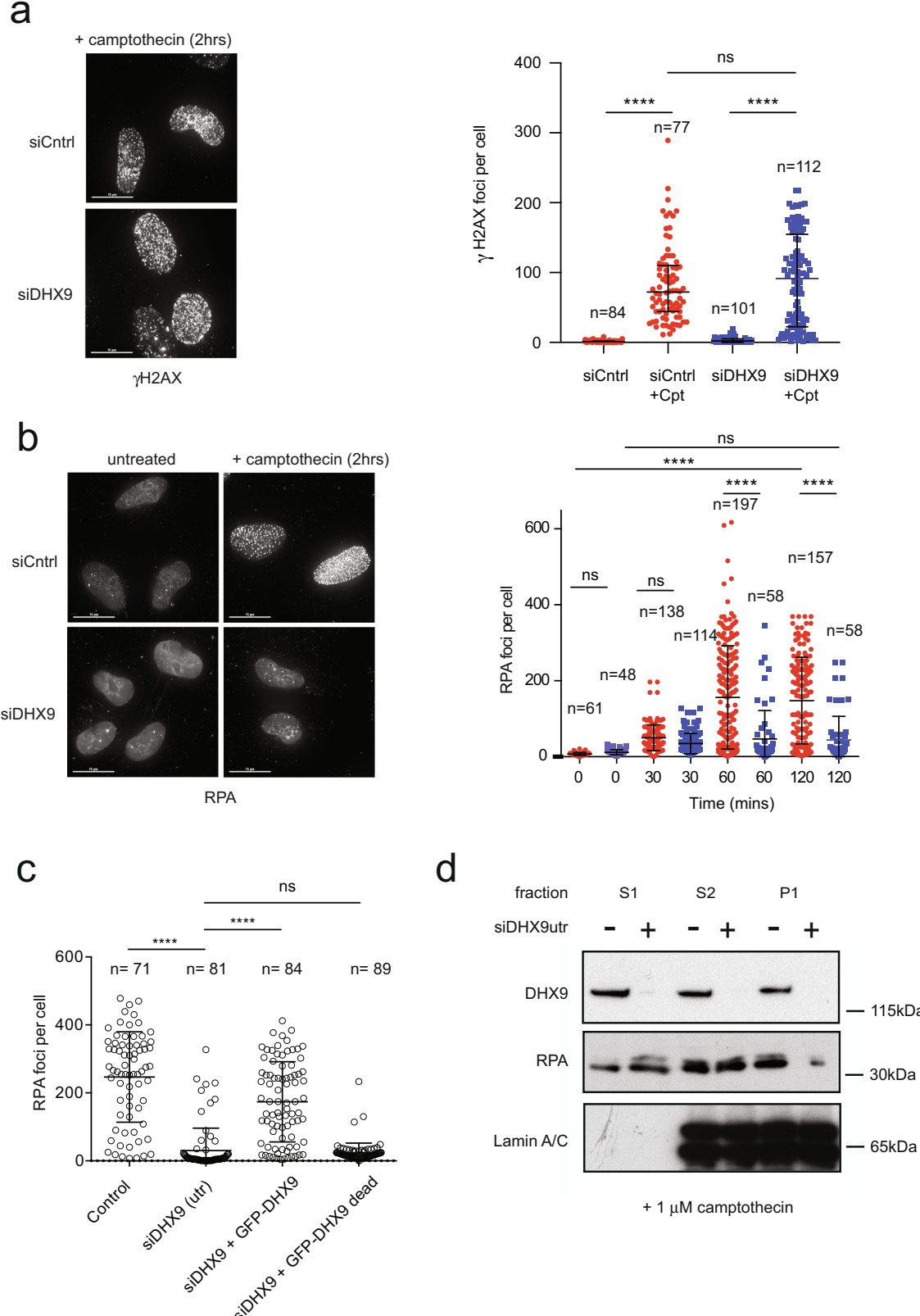

associate with RNA Pol II holoenzyme as a single preformed complex and that both proteins are required for its binding to RNA. This hypothesis was supported by fluorescence images showing that the recruitment of BRCA1 to Cpt-induced DNA damage foci was diminished in cells knocked down for DHX9 (Fig. 7c). Moreover, the recruitment of the complex was

dependent on the unwinding activity of DHX9 since overexpression wild type but not helicase dead DHX9 mutant protein partially restored BRCA1 foci (Supplementary Fig. 5b). The accumulation of BRCA1 in damage-induced foci, like those of DHX9, was greatly also diminished in cells treated with RNase A indicating that recruitment of BRCA1 in Cpt-induced foci is

**Fig. 4 DHX9 is required in DNA end-resection. a** Fluorescent image (left panel) with quantification (right panel) showing similar levels of γH2AX foci in siControl (red) and siDHX9 (blue) cells treated with 1 μM camptothecin for 2 h. **b** Fluorescent image (left panel) with quantification (right panel) showing formation of RPA foci in siControl (red) and siDHX9 (blue) cells treated with 1 μM camptothecin for 2 h then recovered without the drug for the indicated times. DHX9 defective cells are impaired in formation of RPA foci. **c** RPA foci are dependent on the helicase activity of DHX9 and are diminished in cells overexpressing siRNA resistant helicase GFP-DHX9 dead mutant. Cells were treated with 1 μM camptothecin for 2 h. Quantification of *n* cells (as indicated) from three pooled biologically independent experiments were performed in (**a**), (**b**), and (**c**). Mean and error bars for one standard deviation are shown. Data sets were shown to be significantly different using a one-way ANOVA test with Tukey's post hoc analysis as described in Methods. (ns not significant, *$p < 0.1$, **$p < 0.01$, ***$p < 0.001$, and ****$p < 0.0001$). **d** Western blot showing reduced RPA in the chromatin (P1) fraction of cells treated with 1 μM camptothecin (2 h) and knocked down for DHX9. RPA in the cytoplasmic (S1) and nucleoplasm (S2) fractions is not affected by the knockdown of DHX9. Lamin A/C is shown as a control for nuclear fractionation. Source data are provided as a Source Data file.

dependent on RNA (Fig. 7c). Lastly, the recruitment of DHX9 and BRCA1 in nuclear foci was suppressed by treating cells with the transcription inhibitor DRB (5,6-dichloro-1-β-D-ribofuranosylbenzimidazole) (Supplementary Fig. 5c).

Therefore, our data support a model in which BRCA1 and DHX9 are recruited to sites of DNA damage as a preformed complex that binds to nascent RNA generated by RNA Pol II-mediated transcription where they facilitate the repair of DNA damage by HR.

**DHX9 contributes mechanistically to HR.** Previous studies established that the HR defect in BRCA1-deficient cells is suppressed by knockdown or mutation of the end-joining proteins 53BP1 and RIF1[27–29]. While DNA end resection is severely impaired in BRCA1 defective cells, it occurs at almost normal levels in $BRCA1^{-/-}53BP1^{-/-}$ cells[28]. Similarly, cells that are defective in BRCA1 alone are hypersensitive to Olaparib, while cells deficient in both BRCA1 and 53BP1 are resistant. Therefore, while BRCA1 is an important mediator of the HR pathway, it does not perform a mechanistic role in HR-mediated DSB repair. Instead, BRCA1 channels DSB for HR repair by suppressing 53BP1-mediated DNA end-joining. This process is referred to as pathway choice.

The suppression of NHEJ by BRCA1 is achieved, at least in part, by its ability to inhibit the recruitment of RIF1 to sites of DNA damage[29]. Therefore, we looked to see whether the same was true for DHX9. We first confirmed that the treatment of cells with IR resulted in the recruitment of RIF1 to punctate nuclear foci. Although RIF1 was also recruited to chromatin after treatment of cells with Cpt, this staining was less punctate, most likely reflecting the different mechanism through which Cpt and IR induce DNA damage. Nevertheless, depletion of DHX9 led to an increased accumulation of RIF1 on chromatin in response to Cpt (Fig. 8a). This was consistent with our earlier finding that cells that are deficient in DHX9 exhibit increased levels of DSB repair by NHEJ. Importantly, it confirmed that like BRCA1, DHX9 suppresses the recruitment of Rif1 to chromatin during pathway choice.

If the sole function of DHX9 in HR is to facilitate recruitment of BRCA1 to inhibit 53BP1-mediated end joining, one might expect that it would be dispensable for HR in cells simultaneously knocked down for 53BP1. However, this was not the case (Fig. 8). Simultaneous knockdown of DHX9 and 53BP1 caused only a small increase in the generation of RPA foci (Fig. 8b) and a partial suppression of HR in a pDR-GFP DSB repair assay (Fig. 3c). siDHX9si53BP1 double knockdown cells also remained hypersensitive to Olaparib treatment (Fig. 8c). This indicated that, in addition to its role in pathway choice, DHX9 also contributes mechanistically to HR.

What then might this role be? One possibility was that DHX9 promoted the recruitment and function of proteins that are directly involved in DNA resection. To address this we treated cells with Cpt and measured the recruitment of the resection

proteins CTIP and BLM to chromatin. Interestingly, in cells depleted of DHX9, recruitment of both CTIP and BLM to nuclear foci in response to Cpt was diminished (Fig. 9a, b). Consistent with this observation, CTIP and BLM were reduced in the chromatin fraction of cell extracts prepared from DHX9 depleted cells (Fig. 9c). From this, we concluded that, in addition to its role in pathway choice, DHX9 enhanced the recruitment and/or stability of CTIP and BLM on chromatin during DNA end resection.

**DHX9 is required for DNA damage-induced replication restart.** In the absence of pathway choice, CTIP mediated DNA resection can occur independently of BRCA1. However, we observed that the defect in resection of Cpt-induced DSB in both BRCA1 defective and DHX9 defective cells was only partially suppressed by knockdown of 53BP1. Previously, BRCA1 was hypothesized to play a role in the processing of DNA ends that are blocked and, therefore, refractory to DNA replication. These might include DNA ends terminating in top Icc or ends containing chain terminating nucleotides[30,31]. In the absence of BRCA1, these blocked DNA ends are less permissive for the restart of DNA synthesis that is critical for their repair. Mohiuddin et al.[30] demonstrated further that resumption of DNA synthesis for DNA ends containing chain-terminating nucleotides required a pathway that was dependent on the interaction of BRCA1 with CtIP. Importantly this function of BRCA1 was independent of 53BP1 and pathway choice.

Since DHX9 delivers BRCA1 to Cpt-induced DSB, we predicted that it might also contribute to the processing of top Icc end intermediates to facilitate DNA replication. To address this we modified the assay of Mohiuddin et al.[30] for Cpt-induced DSB. We incubated cells for 30 min in media containing the nucleotide analog CldU to enable its incorporation into the genomes of replicating cells. CldU was washed out from the media and the cells were incubated for 2 h in fresh media containing 5 μM Cpt to induce DSB. After removal of the drug, we incubated cells for a further 1 h in fresh media containing a second nucleotide analog, IdU. The incorporation of these nucleotides into DNA was detected using specific antibodies and visualized by fluorescence imaging. Cells that were undergoing DNA replication before Cpt treatment were identified by the incorporation of CldU into DNA. Cells that were permissive for DNA replication after Cpt treatment were identified by the additional incorporation of IdU.

Wild-type cells were proficient for replication restart after Cpt treatment and therefore incorporated both IdU (green) and CldU (red), appearing mainly yellow when images were merged. As expected, cells knocked down for BRCA1 were impaired in replication restart and therefore exhibited fewer yellow cells and more cells incorporating IdU alone (green) (Fig. 9d). Importantly, we confirmed that this defect was not suppressed by the simultaneous knockdown of 53BP1 and was therefore independent of pathway choice. Critically, the same was true for cells

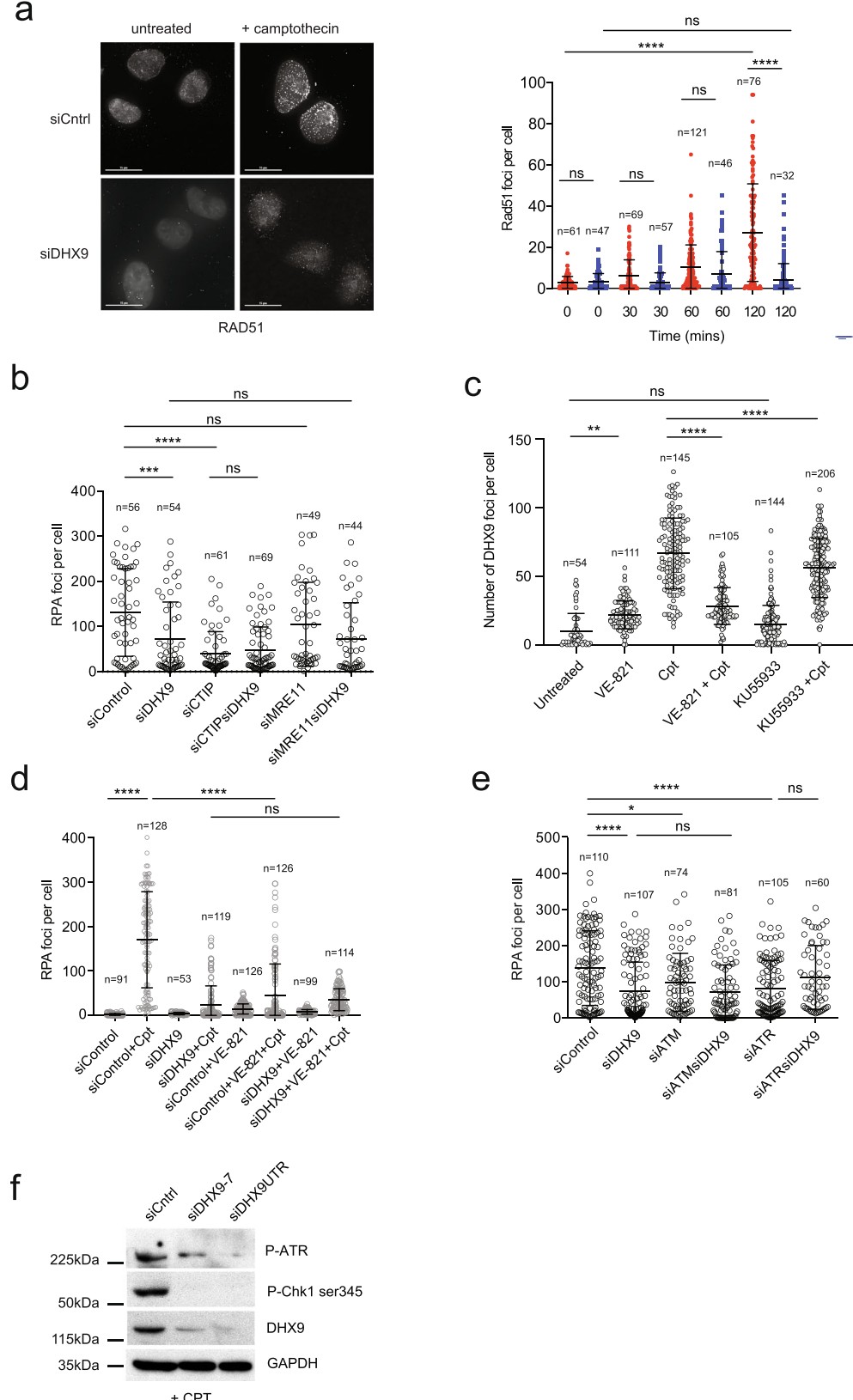

knocked down for DHX9. Significantly fewer of these cells were competent for DNA synthesis after treatment with Cpt and therefore primarily stained green. This defect was not suppressed by a loss of 53BP1. From this, we concluded that DHX9 plays an important role in the restart of DNA synthesis in response to Cpt-induced DNA damage. We hypothesize that DHX9, along

with BRCA1 and CTIP, processes top Icc blocked DNA ends making them permissive for DNA end resection and for repair synthesis in a reaction that is independent of pathway choice.

Together, our data argue that DHX9 is a key component in the DDR, linking transcription with the processing and resection of broken DNA ends for HR.

**Fig. 5 DHX9 is required for the recruitment of RAD51. a** Fluorescent image (left panel) and quantification (right panel) showing the decreased formation of RAD51 foci in control U2OS cells (red) and cells knocked down for DHX9 (blue) treated with 1 μM camptothecin for 2 h then left for 2 h to recover. **b** DHX9 is in a common genetic pathway for the formation of RPA foci with CTIP and MRE11. RPA foci in cells knocked down using siRNA for the indicated genes are shown. Cells were treated with 1 μM camptothecin for 2 h. **c** Recruitment of DHX9 to foci is dependent on ATM and ATR. Wild-type cells were inhibited for ATR and ATM using VE-821 and KU55933, respectively. **d, e** DHX9 is in a common genetic pathway for the formation of RPA foci with ATR and ATM. RPA foci are shown for cells treated with inhibitors for ATR (VE-821) and ATM (KU55933) (**d**) or knocked down with siRNA against ATM and ATR (**e**). **f** Western blot showing that knockdown of DHX9 impairs Cpt induced autophosphorylation of ATR on Thr1989 and phosphorylation of Chk1 on ser 345. Quantification of *n* cells (as indicated) from three pooled biologically independent experiments were performed in (**a–e**). Means of data sets were shown to be significantly different using one-way ANOVA with Tukey's post hoc test. (ns not significant, *$p < 0.1$, **$p < 0.01$, ***$p < 0.001$, and ****$p < 0.0001$). Error bars indicating one standard deviation are also indicated. Source data are provided as a Source Data file.

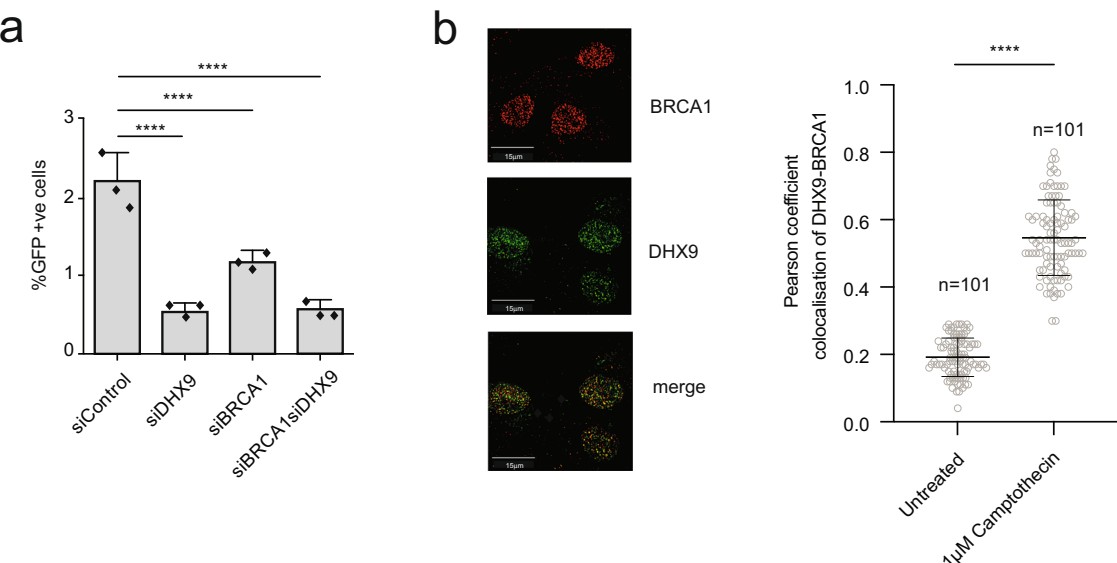

**Fig. 6 DHX9 and BRCA1 function in the same genetic pathway. a** BRCA1 and DHX9 are in the same genetic pathway for the repair of an *I-Sce1* induced DSB by HR measured in a pDR-GFP assay. Data were from three independent experiments. Statistical significance was established using a one-way ANOVA with post hoc Tukey's test (****$p < 0.0001$). **b** Fluorescence images of BRCA1 (red) and DHX9 (green) co-localize in DNA Damage induced nuclear foci. The graph depicts the Pearson coefficient for co-localization of DHX9 and BRCA1 in untreated cells and cells treated with 1 μM camptothecin for 2 h. Differences in mean values for three independent experiments were shown to be statistically significant using a one-way *t*-test (****$p < 0.0001$). Error bars indicating one standard deviation are also indicated. Source data are provided as a Source Data file.

## Discussion

Current understanding of how DHX9 contributes to the maintenance of genomic instability has focussed on its ability to unwind unusually stable nucleic acid secondary structures that occur throughout the human genome[6,12,13]. We show here that DHX9 also plays a critical and direct role in the repair of DSB by HR. We demonstrate that DHX9 functions early in HR to promote the processing and resection of broken DNA ends, generating the 3′ ssDNA substrate for RAD51 recombinase to drive HR. Moreover, we establish that this function is dependent on the direct physical interaction of DHX9 with BRCA1 and that, together, these proteins associate with the RNA Pol II holoenzyme mediated through RNA. Our findings identify DHX9 as a key component of the DDR, linking RNA processing with the repair of DNA breaks.

Like many DDR proteins, DHX9 accumulates at sites of DNA damage. DHX9 accumulates efficiently to DSB induced by treating cells with Cpt but is recruited to IR induced DSB much less frequently. Accordingly, cells that are defective for DHX9 are hypersensitive to treatment with Cpt, but not to IR. This suggests that DHX9 is recruited to and is required for the repair of a specific subset of DSB, rather than all breaks. What then defines this group of breaks? The majority of breaks generated by Cpt-induced inhibition of top I are thought to arise through transcription-replication collisions (TRC) that occur when DNA replication encounters trapped or stalled RNA Pol II complexes. These TRC are reasonably common but are greatly elevated in precancerous and tumor cells.

Since DHX9 is a component of the RNA Pol II holoenzyme we speculate that it primarily accumulates at DSB that occur in transcribed regions of the genome. This is supported by our data showing that the recruitment of DHX9 to Cpt-induced foci is dependent on both transcription and on RNA. While IR-induced DSB may also occur at transcribed loci, these breaks are generated by direct scission of DNA strands and not by TRC. Consequently, IR-induced DSB are more likely to comprise canonical two-ended DSB that are an efficient substrate for repair by NHEJ. Collapsed replication forks arising from TRC, by contrast, are more likely to generate SEB that requires homologous recombination for their repair.

This link to transcription also explains why Cpt-induced DHX9 foci become more intense over time. We speculate that DNA breaks in regions of high transcription will lead, over time, to an accumulation of RNA Pol II and therefore also DHX9 at the break site. Accordingly, persistent DSB become more intensely stained for DHX9. Interestingly, the accumulation of DHX9 at DSB also coincides with the generation of DNA-RNA hybrid. This is consistent with our previous study showing that DHX9 promotes the generation of R-loops at sites of stalled or trapped RNA Pol II and argues against the alternative hypothesis that

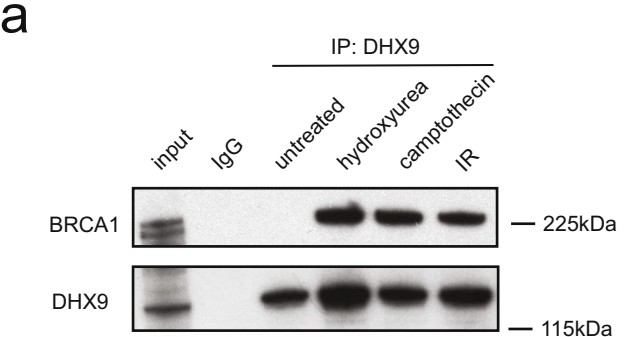

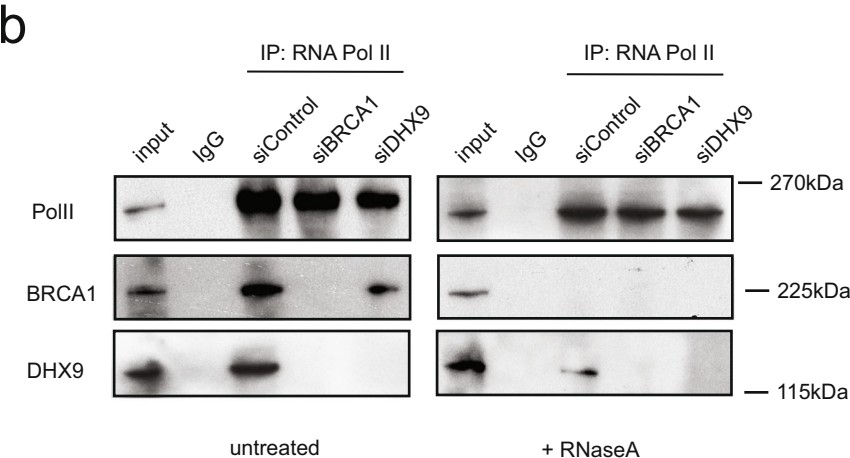

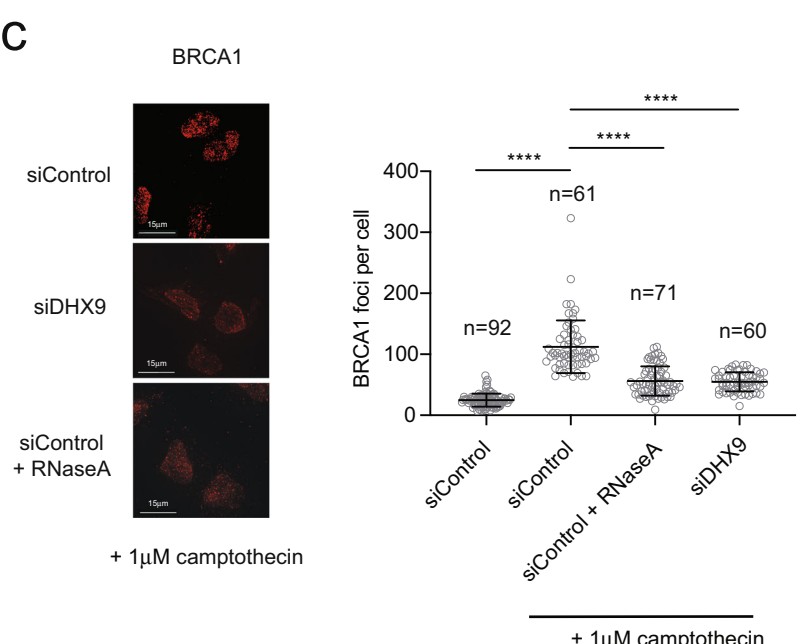

DHX9 removes R-loops. Nevertheless, it is currently unclear whether these R-loops are simply by-products of transcriptional pausing or whether they are generated as obligate intermediates in DSB repair.

Rapid repair of DNA damage at expressed loci is critical for the preservation of normal cell function. A paradigm for the preferential removal of DNA damage from expressed genes exists for transcription-coupled nucleotide excision repair[32]. In cells, a similar coupling of transcription with DSB repair would confer significant benefits for the maintenance of genomic stability. In support of such a pathway, Aymard et al. reported that DSB located in transcribed loci are preferentially repaired by RAD51-dependent HR, while DSB in silent chromatin showed no preference for HR over NHEJ[1]. Moreover, Yasuhara et al. proposed a

**Fig. 7 BRCA1 and DHX9 form a complex that interacts with RNA Pol II-mediated through RNA. a** Western blot showing co-immunoprecipitation of BRCA1 with DHX9 in cells treated with DNA damaging agents; 2 mM hydroxyurea (for 4 h), 2 μM camptothecin (for 2 h), and ionizing radiation (10 Gy) (as indicated). All samples were treated with RNaseA to exclude interactions mediated by RNA. **b** Western blot of RNA Pol II immunoprecipitation from U2OS cells knocked down for specific genes using siRNAs indicated and treated with 1 μM camptothecin for 2 h (all samples). Blot shows co-purification of BRCA1 and DHX9 with RNA Pol II is dependent on the presence of both DHX9 and BRCA1 in cells. Co-precipitation of DHX9 and BRCA1 with RNA Pol II is also dependent on RNA as the interaction is disrupted in samples treated with RNaseA (indicated). **c** Fluorescence image (left panel) with quantification (right panel) showing that DNA damage-induced BRCA1 foci are diminished in cells knocked down for DHX9 and in cells treated with RNaseA. Graph (right panel) shows the quantification of *n* cells (as indicated) from three pooled biologically independent experiments. Mean values were shown to be significantly different using one-way ANOVA with post hoc Tukey's test (****$p < 0.0001$). Mean and standard deviation are shown. Source data are provided as a Source Data file.

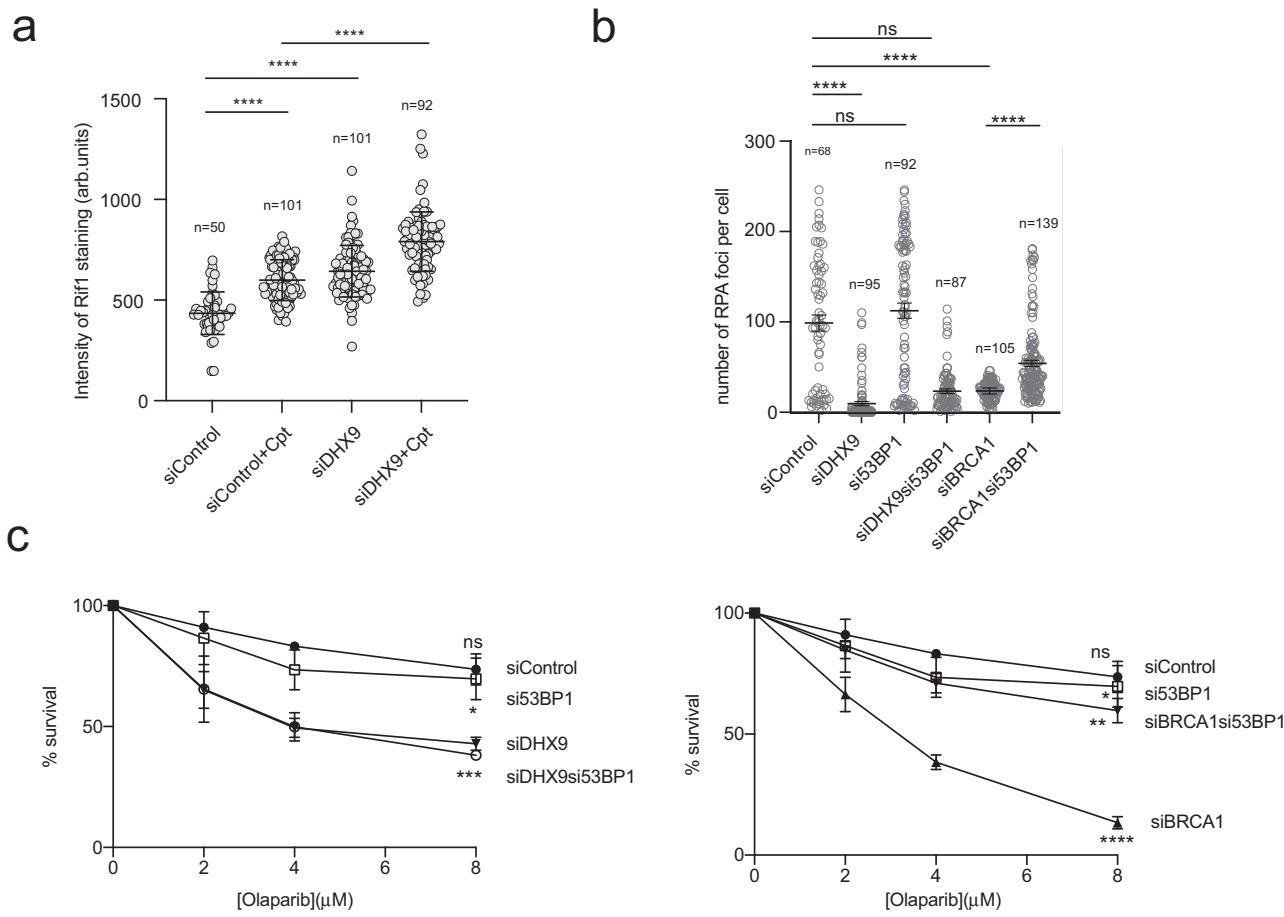

**Fig. 8 Defects in DHX9 are not suppressed by knockdown of 53BP1. a** DHX9 suppresses the recruitment of RIF1 to chromatin in response to camptothecin-induced DNA damage. The intensity of chromatin-bound RIF1 staining is plotted. Mean and standard deviation are indicated. **b** Defect in the recruitment of RPA into foci in DHX9 defective cells is only partially restored by knockdown of 53BP1. A number of RPA foci is plotted for cells knocked down for the indicated genes using siRNA. Quantification of *n* cells (as indicated) from three pooled biologically independent experiments were performed in (**a**) and (**b**). **c** Knockdown of 53BP1 does not restore Olaparib resistance to DHX9 depleted cells in a clonogenic survival assay (left panel) but does restore Olaparib resistance to BRCA1 defective cells (right panel). The data sets for DHX9 and BRCA1 were performed concurrently, with the same controls, but are depicted separately for presentation purposes. Survival values were quantified from *n* = 3 biologically independent experiments. (**a–c**) Statistical analysis performed using one-way ANOVA and multiple comparisons analysed using post hoc Tukey's test in (**a**) and (**b**). (ns not significant, *$p < 0.1$, **$p < 0.01$, ***$p < 0.001$, and ****$p < 0.0001$). Means and error bars indicating one standard deviation are also indicated. Source data are provided as a Source Data file.

model for transcription-associated homologous recombination repair (TA-HRR) in which R loops recruit RAD52 and BRCA1 to initiate HR and suppress recruitment of RIF1-53BP1 for NHEJ[33].

Notwithstanding these and other models, the pivotal roles of DHX9 in RNA Pol II-dependent transcription and HR suggest it is likely to play a key role in any pathway for transcription-associated DSB repair. Central to its HR function is the association of DHX9 with BRCA1. Previous studies reported that

BRCA1 and DHX9 interact even in the absence of DNA damage[26]. However, we detected only a low-level interaction in unperturbed cells. In contrast, the formation of the BRCA1-DHX9 complex was significantly increased in cells exposed to various DNA damaging agents, including IR. One possibility is that in the original nonphysiological experiments, overexpression of individual domains of BRCA1 and DHX9 in yeast two-hybrid and GST pull-down experiments alleviated the requirement for

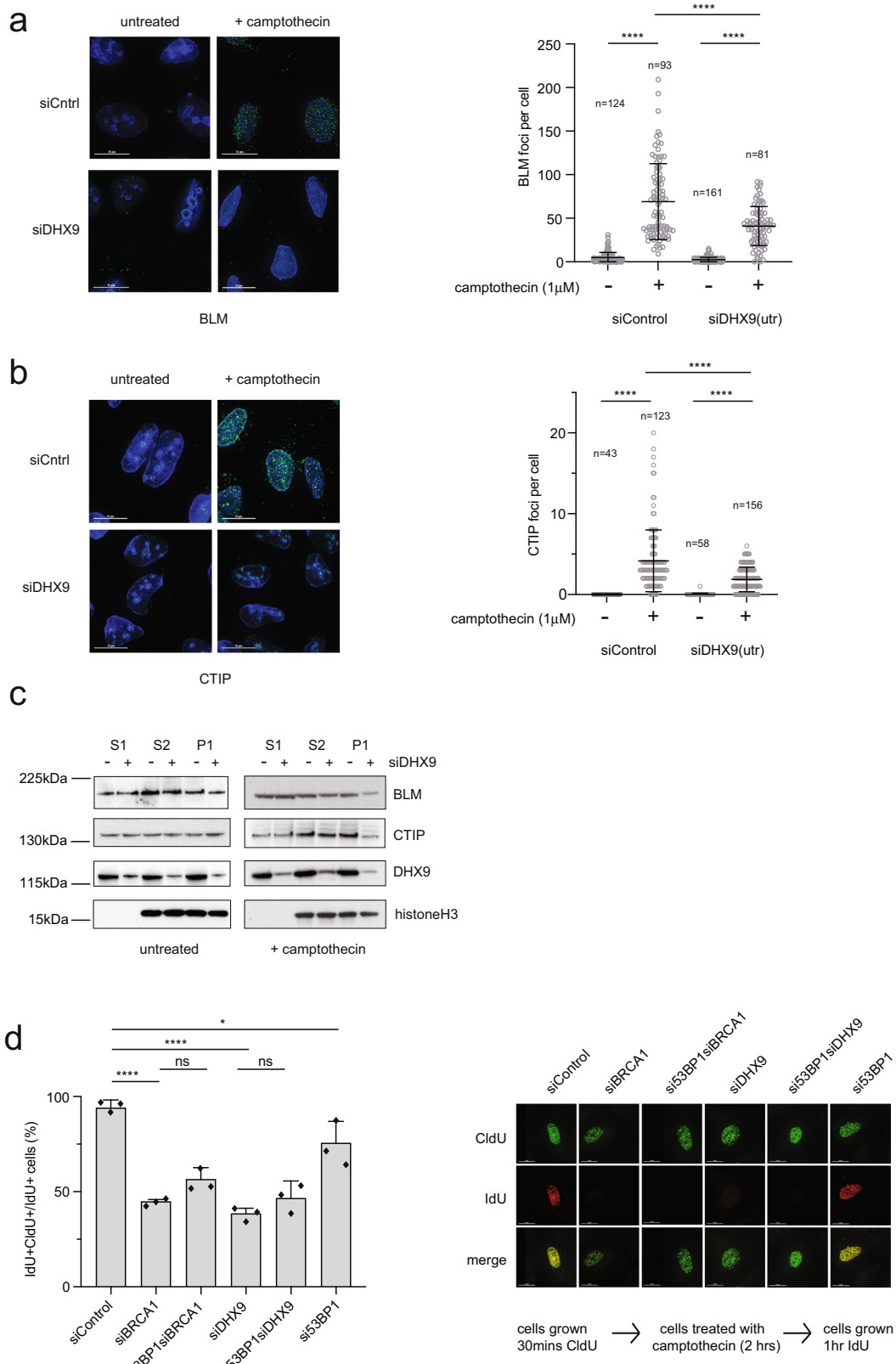

DNA damage. Alternatively, it is possible that DHX9 and BRCA1 interact constitutively, but that the complex is enriched or stabilized in cells with high levels of stalled of RNA Pol II, enhancing its detection.

The interaction of DHX9 with BRCA1 is mediated through the carboxy-terminal region of BRCA1 containing the BRCT domains[26]. A number of other important complexes are formed by proteins that bind to the BRCT domains of BRCA1, including several that are implicated in the resection of DSB. These include ABRAXAS, BRIP1, and CTIP proteins that bind the BRCT domains of BRCA1 to form the BRCA1-A[34], BRCA1-B[35], and BRCA1-C[36] complexes, respectively. Consistent with the previous

**Fig. 9 DHX9 promotes the recruitment of CTIP and BLM to DNA damage. a** Fluorescence images (left panel) and graph (right panel) showing that localization of BLM to camptothecin-induced DNA damage foci is impaired in cells knocked down for DHX9. **b** Fluorescence images (left panel) and graph (right panel) showing that localization of CTIP to camptothecin-induced DNA damage foci is impaired in cells knocked down for DHX9. Quantification of *n* cells (as indicated) from three pooled biologically independent experiments were performed in (**a**) and (**b**). Means were shown to be significantly different using one-way ANOVA with post hoc Tukey's test (****$p < 0.0001$). **c** Western blot of fractionated cell extracts showing that localization of BLM and CTIP to chromatin (P1 fraction) in response to camptothecin-induced DNA damage is reduced in cells knocked down for DHX9. Localization of BLM and CTIP in cytoplasmic (S1) and nuclear fractions (S2) is not decreased. Histone H3 is shown as a marker of S2 and P1 fractions. **d** DNA synthesis is impaired in DHX9 and BRCA1 deficient cells treated with camptothecin (5 $\mu$M for 2 h). This defect is not suppressed by knockdown of 53BP1 Right panel shows representative images for the incorporation of CldU and IdU nucleotide analogs as well as merged images. The left panel shows graphical data of cells stained with both CldU and IldU as a percentage of total cells stained with CldU. Graphs include data from three biologically independent experiments. Mean and error bars indicating one standard deviation are also indicated. Statistical significance for all experiments was demonstrated using one-way ANOVA with post hoc Tukey's test (****$p < 0.0001$, *$p < 0.1$, ns not significant). Source data are provided as a Source Data file.

nomenclature on BRCT domain-interacting complexes, BRCA1-DHX9 might be named BRCA1-D. Unfortunately, however, this name has been used to describe the interaction of BRCA1 in a complex with PALB2, BRCA2, and KEAP1[37]. Therefore, we subsequently refer to BRCA1-DHX9 as BRCA1-DX.

The formation of the BRCA1-DX complex is essential for its recruitment to RNA Pol II and for its role in HR. DHX9 protein was previously reported to act as a bridge for the interaction of BRCA1 with RNA Pol II through direct protein–protein contacts[8,26]. Our data reveal a critical role for RNA in the recruitment of BRCA1-DX to RNA Pol II with only a small contribution from protein–protein interactions. This explains why DNA damage-induced BRCA1 and DHX9 foci are sensitive to treatment with RNaseA. In vitro, BRCA1 binds weakly to DNA[38,39] whereas DHX9 binds to both RNA and DNA. In cells, DHX9 did not associate with RNA Pol II in the absence of BRCA1. Hence, BRCA1 and DHX9 are incorporated into the RNA Pol II holoenzyme as a preformed complex that binds nascent RNA. Intriguingly, this raises the possibility that BRCA1 contributes to other functions involving DHX9, for example during co-transcriptional processing of pre-mRNA. This hypothesis is supported by a recent report where overexpression of the RNA splicing associated oncogene EWS-FLI1, a known interactor of DHX9, caused BRCA1 to become trapped on chromatin with RNA Pol II in the absence of exogenous DNA damage[40]. Interestingly these cells are also defective in HR, highlighting the potential of DHX9 to link transcription with the repair of DNA damage by HR.

Our data suggest a model where RNA Pol II delivers BRCA1-DX to DNA damage to promote DNA end resection and recruitment of RPA to DNA (Supplementary Fig. 5d). Previous studies showed that BRCA1 facilitates end resection by stimulating the activity of CTIP[41,42]. This is consistent with our genetic data showing that DHX9 and CTIP operate in a common pathway that responds to the DNA damage signaling kinases ATR and ATM. Accordingly, we demonstrated that the stable recruitment of both CTIP and BLM to damaged chromatin is less efficient in the absence of DHX9. We speculate that, by recruiting BRCA1 to DSB, DHX9 promotes CTIP function and facilitates the formation of the BRCA1-C complex, which together with MRN initiates DNA end-resection. Likewise, the ability of DHX9 to enhance the recruitment of BLM might increase the processivity of DNA resection. Subsequent turnover of BRCA1-C and the formation of the BRCA1-A complex is thought to modulate the overall extent of this resection[43,44]. Hence the sequential assembly of these interrelated complexes is potentially an important mechanism to regulate resection and the repair of DSB by HR. How the formation and resolution of various BRCA1 complexes are orchestrated will be important to establish in the future.

The relationship between DHX9 function and DNA damage signaling is intriguing. Our data suggest that DHX9 responds to

damage signaling by the ATR kinase and less so to ATM. This is not surprising since Cpt-induced DSB are likely generated by replication stress, which is a known substrate for ATR signaling. Depletion of either ATR or DHX9 result in severely decreased end resection and defective HR. However, our data suggest that it is not the case that ATR simply signals for downstream activation of DHX9 in DSB repair. DHX9 also stimulates autophosphorylation of ATR that promotes damage signaling. In the absence of DHX9, we detected a significant reduction in the phosphorylation of Chk1. Hence the activities of DHX9 and ATR are potentially interdependent. By acting very early in HR, DHX9 might promote the recruitment of other proteins via the transcription complex that are required for ATR activation. Further studies will be required to determine how this might occur.

It is not yet clear whether BRCA1-DX is recruited to RNA Pol II during normal transcription or whether its recruitment involves the expression of DDR RNAs (DDRNA) that are induced in the vicinity of DNA damage, as proposed for recruitment of 53BP1 and MDC1[3]. DDRNAs are thought not to play a role in the initial detection of DNA damage but are needed instead for the secondary recruitment of DDR factors through a mechanism that is currently unknown[45,46]. BRCA1-DX on the other hand has the potential to act very early in the DDR where it could signal the presence of DSB at transcribed loci by coupling the initiation of DNA end resection to the slowing or pausing of transcription complexes as they encounter DNA damage[26]. This is not unprecedented as BRCA1, together with another RNA helicase SENATAXIN (SETX), has been shown to promote the repair of R-loop associated DNA damage at transcriptional pause sites[47]. It is possible, therefore, that BRCA1-DX and BRCA1/SETX are manifestations of the same, or similar, transcription-dependent repair activities.

It is clear that DHX9 plays an important role in HR during pathway choice. Like BRCA1, DHX9 promotes HR by suppressing the recruitment of RIF1 to DNA damage thereby preventing 53BP1 mediated end-joining[29]. However, while both DNA end resection and HR are restored to BRCA1 defective cells in which 53BP1 has been knocked out, this was not the case for DHX9. Instead, siDHX9si53BP1 double knockdown cells were sensitive to Olaparib and impaired for DNA end resection, indicating that the contribution of DHX9 to HR is not limited to the establishment of pathway choice. One potential contribution of DHX9, that lies downstream of pathway choice, is its ability to promote the recruitment of the resection proteins CTIP and BLM as described above. Both proteins are required mechanistically for HR, independent of pathway choice (that is, in cells in which neither BRCA1 nor 53BP1 is functional).

Although, in the absence of pathway choice, CTIP mediated end resection of "naked" DSB occurs independently of BRCA1, the interaction of BRCA1 and CTIP is essential for the repair of DSB with blocked termini. It has been hypothesized that this

includes DNA breaks that are blocked by the presence of Cpt-induced Top Icc[31]. It is likely that processing of Top Icc blocked ends involves endonucleolytic cleavage at or near the DNA terminus by CTIP and/or MRE11 and in contrast to the generation of ssDNA by resection, this activity requires BRCA1. However, the biochemical mechanism for this function has not been established. Our data demonstrate that this pathway also requires DHX9 and raises the question of whether it occurs exclusively at DSB associated with stalled transcription or whether it also operates in the repair of other blocked DNA ends. Nevertheless, the contribution of DHX9 to this function cannot be suppressed by mutation or loss of 53BP1 and is therefore independent of pathway choice. Moreover, it highlights the multifunctional contribution of DHX9 in HR.

Our understanding of how RNA and transcription contributes to the repair of DNA damage is still emerging and a subject of great interest. The present work adds DHX9 to a growing number of RBP that have been implicated in the maintenance of genome stability and the repair of DNA breaks. Importantly, we have shown how DHX9 links RNA and RNA Pol II to the repair of DSB and the restart of DNA replication through the formation of the BRCA1-DX complex. Our demonstration that BRCA1-DX, like BRCA1-A and BRCA1-C, plays a pivotal role in DNA end resection highlights the importance of this step in the regulation of DSB repair and uncovers a key contribution played by RNA in the repair of DNA damage by HR.

## Methods

**Cell culture**. HeLa and U2OS cell lines were cultured in Dulbecco's Modified Eagle's Medium (DMEM) supplemented with 10% fetal bovine serum (FBS) and 5% ampicillin. HeLa and U2OS cell lines were obtained from ATCC, H1299-dA3-1 was a kind gift of Professor Takashi Kohno (National Cancer Center Research Institute. Tokyo, Japan)

**Antibodies, chemicals, and reagents**. Primary antibodies used in this study and concentrations used for immunoprecipitatation (IP), immunofluorescence (IF), and Western blot (WB) are indicated. Rabbit polyclonal anti-RNA Helicase A (ab26271, Abcam, for IP 5 μg/ml, IF 1:1000, and WB 1:2000 dilution), mouse monoclonal anti-BRCA1 antibody (OP92, Ab-1; Calbiochem, for IPs 10 μg/ml, IF 1:250, and WB 1:200 dilution), mouse anti-RPA32 (RPA2 Ab #2; Calbiochem, IF 1:500, WB 1:1000, and for FACS 1:100 dilution), rabbit polyclonal anti-RNA-Polymerase II (N-20; Santa Cruz Biotechnology, IP 4 μg/ml and WB 1:1000 dilution), mouse monoclonal anti-GAPDH antibody (GT239; GeneTex, WB 1:1000 dilution), mouse monoclonal anti-phospho-Histone H2A.X (clone JBW301; EMD Millipore, IF 1:250 dilution and WB 1:500 dilution), rabbit anti-phospho-Histone H2A.X (ab81299, Abcam, IF 1:1000 dilution), rabbit polyclonal anti-Rad51 antibody (H-92; Santa Cruz Biotechnology, IF 1:250 dilution), rabbit polyclonal anti-53BP1 Antibody (NB100-904; Novus Biologicals, IF 1:500, WB 1:2000), rabbit polyclonal anti-53BP1 Antibody (A300-272A, Universal biologicals, IF 1:1000 dilution), mouse monoclonal Anti-BrdU antibody (B44; BD Biosciences, 1:1000 dilution for IF), Rat monoclonal anti-BrdU antibody (ab6326, Abcam, IF 1:1000 dilution), rabbit polyclonal anti-CTIP antibody (NB100-79810, Novus Biologicals, WB 1:2000, IF 1:500 dilution), rabbit polyclonal anti-MRE 11 antibody (NB100-142, Novus Biologicals, WB 1:1000 dilution), rabbit polyclonal anti-BLM antibody, (PLA0029, Sigma-Aldrich, WB 1:1000 dilution), mouse monoclonal anti-ATR antibody (NB100-308, Novus Biologicals,WB 1:1000 dilution), rabbit polyclonal anti-ATM antibody (NB100-104, Novus Biologicals, WB 1:1000 dilution), rabbit monoclonal anti-Ku80 Antibody (218, Cell Signaling, WB 1:1000 dilution). Secondary antibodies: goat anti-mouse Alexa Fluor 488 and 647 (Molecular Probes) was used at 1:200 dilution for FACS and 1:1000 dilution for IF, goat anti-rabbit Alexa fluor 488, 568, and 647 (Molecular Probes) was used at 1:1000 dilution for IF, goat anti-rat Alexa 488-conjugated secondary antibody (Molecular Probes, IF 1:600 dilution), goat anti-mouse Alexa 594 conjugated secondary antibody (Molecular Probes, IF 1:500 dilution).

Secondary antibodies for western blot; Peroxidase-AffiniPure Goat Anti-Rabbit IgG (H + L) 111-035-144-JIR (1:5000 dilution) and Mouse IgG antibody (HRP) (GTX213111-01; GeneTex, 1:5000 dilution) were used. IgG controls were Normal Rabbit IgG #2729 (Cell Signaling) and normal mouse IgG (sc-2025; Santa Cruz Biotechnology).

Chemicals; Hydroxyurea H8627 and (S)-(+)-Cpt C9911 (Sigma-Aldrich), AZD 2281- Olaparib (Axon Medchem), 5-Bromo-2′-deoxyuridine (Sigma, B5002), VE-821 (Sigma, SML 1415), RNase A 19101 (17,500 U, Qiagen), Ribonuclease H (Thermofischer, 18021014), Halt™ Protease Inhibitor Cocktail (100X) 78429

(Thermos Scientific) and PhosSTOP™ (Merck), 5-Chloro-2′-deoxyuridine (C6891, Sigma-Aldrich), 5-Iodo-2′-deoxyuridine (17125, Sigma-Aldrich).

**Plasmids**. pMyc-DHX9 was a kind gift of Prof. Jerry Pelletier (McGill University, Canada), pGFP-DHX9 was purchased from Stratech. pMyc-DHX9dead and pGFP-DHX9dead containing D511A and E512A mutations were made by site-directed mutagenesis using Q5 site-directed mutagenesis system (New England Biolabs) according to the manufacturer's instructions.

**SiRNA**. The siRNAs were purchased from Dharmacon-ON-TARGET plus. Non-targeting siRNA Control- D-001810-01-05; DHX9- J-009950-07, DHX9utr- CTM-310164, and CTM-478066; 53BP1- J-003548-07; BRCA1- J-003461-09, ATR- L-003202-00, ATM- L-0030201-00, CTIP- J-011376-07, MRE11A- J-009271-07, and XRCC5 (Ku86)- J-010491-08-0005. SiRNA mediated knockdown of genes was performed using Lipofectamine RNA MAX (Invitrogen) according to the manufacturer's instructions. Knockdown of different proteins was confirmed by western blot (Supplementary Fig. 6).

**Homologous recombination assay by FACS**. About 75,000 U2OS cells containing the pDR-GFP reporter gene were plated in a six-well plate overnight and transfected with siRNA. After 48 h 2 μg of the I-Sce1 expression vector pCBASce was transfected into cells. After a further 48 h cells were harvested and the number of GFP-positive cells was measured by FACS (Fortessa- BD Biosciences). For complementation experiments, a plasmid expressing a wild type of mutant DHX9 was transfected simultaneously with incubation of siRNA.

**Nonhomologous end-joining assay by FACS**. H1299dA3-1 cells were knocked down for indicated genes using siRNA as above and grown for 48 ho. About 50,000 cells were seeded in a 12-well plate and transfected with 1 μg pCBASce plasmid DNA using X-tremeGENE™ HP DNA Transfection Reagent (XTGHP-RO, Roche) and grown for a further 48 h. Cells were harvested by trypsinization, washed with PBS, and GFP expressing cells quantified using FACS (Fortessa cytometer- BD Biosciences). All experiments were performed as three or more biological replicates.

**Immunoprecipitation**. Prior to immunoprecipitation, the primary antibody was incubated with Dynabeads protein G beads (Invitrogen) overnight at 4 °C. Cells were lysed using Lysis Buffer (150 mM NaCl, 0.5% Triton X-100, 50 mM Tris-HCl pH 7.5, NaCl, 1 mM EDTA, 5% glycerol, and 0.1% SDS) supplemented with protease and phosphatase inhibitor cocktails (Thermo Scientific). Lysed cells were passed through a 23 G needle ten times on ice and incubated for 30 min at 4 °C with shaking. For BRCA1, 500 mM NaCl was included in the lysis buffer. Where indicated 3 μl of RNaseA was added to 1 ml of extract and incubated at 37 °C for 30 min. The supernatant was cleared by high-speed centrifugation at 16,000x*g* at 4 °C in a benchtop microcentrifuge. The cleared lysate (1 mg) was added to the antibody-Dynabead complex and incubated at 4 °C with rotation for between 4 and 12 h. Immunocomplexes were separated using a magnet, washed three times in lysis buffer, boiled in sample buffer, and loaded on a 4–12% Bis-Tris poly-acrylamide gel (Invitrogen). Proteins were transferred to the PVDF membrane using a Novex transfer system (Invitrogen) and immunoblotted using the indicated antibodies.

**Fluorescence imaging**. U2OS cells were seeded on coverslip overnight, non-chromatin bound proteins were pre-extracted using 0.5% TritonX-100 CSK buffer (25 mM Hepes pH 7.4, 50 mM NaCl, 1 mM EDTA, 3 mM MgCl₂, 300 mM sucrose) for 2–5 min on ice and then fixed with 3.7% paraformaldehyde. For RNase A treatment, cells were either treated for 30 min at room temperature before or after fixation with RNase A (200 μg in 500 μl Phosphate buffered saline (PBS)). Cells were washed three times in PBS and permeabilized for 10 min in 0.5% Triton X-100/PBS. After three additional PBS washes, cells were blocked using 1–3% BSA/PBS for 30 min. Cells were incubated with primary antibody (as indicated above), followed by the addition of Alexa Fluor-488 or -568 or 647 conjugated secondary antibodies (1:1000). To visualize nuclei, cells were also stained with 0.5 μg/ml DAPI (Molecular Probes) for 15 min. Slides were mounted using Prolong gold anti-fade reagent (Invitrogen) and images were acquired using a Delta vision DV4 wide-field deconvolution microscope with a 100X objective. Where indicated, cells were treated with 1 μM Cpt and incubation for various times, after which cells were washed with PBS, pre-permeabilized, fixed, and processed as above.

Cells were seeded on coverslips and grown overnight and then transfected with siRNA for 6 h the media changed and then cells grown for a further 48 h. Where indicated plasmid expressing wild-type DHX9 or D511A E512A helicase "dead" mutant was transfected using lipofectamine 2000 reagent (Invitrogen) for 48 h. Images were analyzed using a Delta Vision microscope and quantified using Image J software.

Image Analysis- Colocalization between X protein and Y protein was assessed in single-cell regions using SoftWoRx (version 5.5.0, release 6). A rectangular selection was defined for each cell in deconvolved images and the Pearson coefficient of correlation was calculated for the volume. A custom-designed macro

for quantifying nuclear foci was developed by the Dundee Imaging Facility. This was used to quantify nuclear foci in two different channels (green and red). Nuclei were automatically detected in the DAPI channel using the Default ImageJ auto-thresholding method and foci in the red and green channels were counted using ImageJ's "Find Maxima" feature with a user-defined Prominence above the background.

**ssDNA detection assay by immunofluorescence.** Cells were grown on microscope slides overnight. siRNA knockdown was performed as above and cells were incubated with 10 μM BrdU 24 h. A total of 1 μM Cpt was added to the media for 2 h and then washed out with PBS. Cells were pre-permeabilized in ice for 5 mins then fixed and treated as above using non-denaturing conditions. Cells were incubated with primary antibody against BrdU overnight then washed and secondary antibody added as above. Images were captured with a Delta vision DV4 wide-field deconvolution microscope using a 40x objective.

**Cell viability assay.** U2OS cells were transfected with siRNA for 48 h as above. About 5000 cells per well were seeded into a 12-well plate (time 0). On days 3 and 7, cells were recovered from the monolayer using trypsin and viable cells counted using Casey Cell Counter. Where indicated Cpt and Olaparib were included in the medium and replenished every 3 days. Cell viability was plotted using data from three to six independent biological replicates.

**Clonogenic survival assay.** Cells were seeded into six-well plates at a density of 1000 cells per well treated with drugs at the indicated concentrations. After incubation for 7 to 10 days, cells were fixed and stained with a 0.05% solution of crystal violet in methanol, containing 1% formaldehyde. The absorbed dye was resolubilized with methanol containing 0.1% SDS for 2 h, 100 μl of which was transferred into 96-well plates and measured photometrically (595 nm) in a microplate reader. Data were representative of three independent experiments.

**Flow cytometry.** Analysis of RPA staining by FACS was performed as previously described with a few modifications[48]. Firstly, non-chromatin bound proteins were extracted with 0.5% Triton X-100 (PBS-T) for 15 min on ice followed by fixation in 4% paraformaldehyde and then permeabilized for 30 min at room temperature. After washing in PBS the cells were incubated with primary and secondary antibody subsequently, washed and stained with 0.02% sodium azide, 250 μg/ml RNase A and 2 μg/ml of 4′,6-diamidino-2-phenylindole (DAPI) in PBS for at least 30 min at 4 °C prior to analysis using a Fortessa flow cytometer (Beckton Dickinson). Data were collected from more than three independent experiments and quantified using Prism v6 (GraphPad Software). Analysis of cell cycle was performed as described previously[6].

**Cell fractionation.** Briefly, $3 \times 10^6$ HeLa cells per condition were collected and suspended in 250 μl of buffer A (10 mM HEPES pH 7.8, 10 mM KCl, 1.5 mM MgCl$_2$, 0.25 M sucrose, 10% glycerol, 1 mM Dithiothreitol (DTT), 0.1% Triton-X-100, protease, and phosphatase inhibitors) and incubated for 5 min on ice. The soluble cytoplasmic fraction (S1) was separated from the nuclei (P2) by centrifugation for 4 min at 1700×g at 4 °C. The nuclear fraction P2 was washed twice with 500 μl buffer A and suspended in 200 μl buffer B (3 mM EDTA, 0.2 mM EGTA, 1 mM DTT, phosphatase, and protease inhibitors) and incubated at 4 °C for 30 min. The insoluble chromatin fraction (P1) was separated from the nuclear soluble proteins (S2) pellet by acid-extraction using 0.25 N HCl and incubated on ice for 30 min. The lysate was then centrifuged at 16,000×g for 15 min at 4 °C. The supernatant (contains acid-soluble proteins) was neutralized using 1 M Tris-HCl pH 8 using 1:5 volume. Twenty-five micrograms of the protein were loaded in 4–12% Bis-Tris gradient polyacrylamide gel and transferred onto the PVDF membrane (Millipore). Membranes were blocked in PBS with 0.1% Tween 20 and probed with respective primary antibodies. Bound proteins were detected with Pierce ECL western blotting substrates and developed with x-ray film (Konica Minolta).

**Replication restart assay.** Hela cells were grown on slides and pulse-labeled for 30 min with 50 μM of CldU. Cells were washed with PBS and then treated with 5 μM of Cpt for 2 hrs, washed with PBS and incubated with fresh medium. Cells were then grown in media containing 50 μM of IdU for 60 mins. Cells were then washed and fixed in 4% paraformaldehyde for 15 min and then permeabilized in 0.5% Triton X-100 for 15 min. Cells were incubated with 2 M HCl for 45 min to denature the DNA and then blocked for 30 min with 5% milk in PBS-T. To visualize CldU, cells were immunostained for 1 h with rat primary monoclonal antibody [BU1/75, Abcam], washed with 0.05% PBS-T 20 and three times for 5 min then immunolabeled for 30 min with goat anti-rat Alexa 488-conjugated secondary antibody (Molecular Probes). To visualize IdU cells were immunostained for 40 mins at 4 °C with mouse anti-BrdU monoclonal antibody (Becton Dickinson). Cells were subsequently labeled for 30 min with goat anti-mouse Alexa 594 conjugated secondary antibody (Molecular Probes) to label IdU. Replication restart is represented by the co-localization of CldU and IdU. Nuclei were counterstained with DAPI for fluorescence imaging.

**Data processing, statistical analysis, and reproducibility.** For scatter graphs, data from *n* cells (as indicated) from three pooled independent experiments were plotted. Mean and error bars representing one standard deviation are shown. Data were analyzed and, where appropriate, the significant differences between the mean values of independent data sets was determined using one-way ANOVA test with ad hoc Tukey's test for multiple comparisons (*$p < 0.1$, **$p < 0.01$, ***$p < 0.001$, ****$p < 0.0001$), as indicated. For these populations, prior to ANOVA, we also established that there were significant differences in the dependent variable using a non-parametric two-tailed Mann–Whitney unpaired *t*-test (not shown). All statistics were performed using Prism v9 (GraphPad Software). Micrographs and western blots represent results from three or greater biologically independent experiments.

**Reporting Summary.** Further information on research design is available in the Nature Research Reporting Summary linked to this article.

## Data availability

Source data are provided with this paper. Any additional reasonable requests for data supporting the findings of this study are available from the corresponding author. Source data are provided with this paper.

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

## Acknowledgements

We thank Professor Jerry Pelletier (McGill University, Quebec, Canada) for the kind gift of a plasmid expressing DHX9-Myc, Professor Takashi Kohno (National Cancer Center Research Institute. Tokyo, Japan) for cell line H1299-dA3-1, Professor Maria Jasin (Memorial Sloane Kettering, New York USA) for plasmid pDR-GFP, and Dr Graeme Ball (Dundee Imaging Facility, University of Dundee) for assistance with image analysis. This manuscript was funded by BBSRC [BB/P021387/1 to K.H.] and by the Ninewells Cancer Campaign.

## Author contributions

P.C. and K.H. contributed to the design, planning, and interpretation of experiments. P. C. performed the experiments and P.C. and K.H. prepared the manuscript.

## Competing interests

The authors declare no competing interests.
