## [Peer Review File · Nature Communications]

REVIEWER COMMENTS

Reviewer #1 (Remarks to the Author):

It has been reported recently that HR is somehow targeted to highly transcribed regions of the genome, while in the absence of transcription, HR is not targeted. In this report, the authors address the question of how HR is targeted to transcribing regions of the genome.

Experiments are conducted, and a critical role for the DHX9 gene is identified in the repair of DSBs by HR in a transcription and RNA dependent manner. The authors conduct a logical series of experiments that have been carefully devised, well controlled and are statistically robust, in terms of what is reported. The authors demonstrate that DHX9 defective cells fail to recruit RPA and RAD51 proteins to DNA damage sites and subsequently are unable to repair DSBs by HR. The experiments described are consistent with the interpretation that DHX9 plays a direct role in the repair of DSBs during HR by recruiting BRCA1 to RNA as a component of the RNA Polymerase II complex in the vicinity of DSB. Importantly, they provide good evidence that DHX9's role in the repair of DSBs is its ability to promote end-resection - an early step in the process of DSB repair by HR.

I have the following suggestions to improve the MS:

1. Given what is being reported about the mechanism, including the key elements that remain to be determined, it would be most helpful to have a current model of the proposed mechanism of DHX9 function in repair of DSBs by HR, as an aide to define future studies.
2. A clearer description of the significance and mechanisms of break induction by the various sources of DSB generating treatments used would be beneficial. Some of the significance of the different break-inducing treatments will not be so clear to the general reader.
3. There is a slight discrepancy regarding whether DHX9 plays a late or early role in the processing and repair of DSBs. The authors eventually conclude that it plays an early role in break resection. However, earlier in the Results section it is suggested that it might have a later role in break repair, based on its break recruitment timing. Alternative explanations are considered by the authors, but this discrepancy should be discussed in more detail to avoid possible confusion.

The manuscript is very well written, with only minor errors that should be eliminated during final editing of the report.

Reviewer #2 (Remarks to the Author):

The manuscript makes a strong case for DHX9 involvement in the resection step of homologous recombination. The data illustrates it is recruited to sites of damage in an RNA -dependent fashion. DHX9 interacts both with BRCA1 and RNA Pol II, indeed, surprisingly its interaction with Pol II depends on the presence of BRCA1. Whether RNA polII : DHX9 :BRCA1 recruits as a unit, as suggested is difficult to gage as only Camptothecin-treated cells have been examined for complex – is it pre-existing or induced?

A controversial and pressing question of how BRCA1 is recruited to DNA double strand breaks has been open for years; the data makes the case that RNA and DHX9 is central to BRCA1 recruitment. BRCA1 recruitment depends both on DHX9, RNA, and surprisingly, the unwinding activity of DHX9. It would be intriguing to understand further why the unwinding activity is required, since interaction between DHX9 and BRCA1 C terminus previously been described, and here RNA and its unwinding is further required.

A good deal of epistatic detail is given between BRCA1 and DHX9, and further ATR epistasis is consistent with a role on long range HR, consistent with BRCA1 regulation. However the fact that 53BP1 cannot rescue RPA foci or Olaparib resistance of DHX9 depleted cells, suggests that the role of DHX9 in HR is additional to BRCA1 recruitment. I'm not sure it's enough not to explore the facile explanations – poor Exo1/ DNA2:BLM recruitment, an examination of 53BP1-Rif1, are they as expected?

A positive control for Fig 8 a and b in which RPA foci and Olaparib resistance are restored in BRCA1 deficient cells by 53BP1 siRNA is needed (a facile explanation it that the siRNA doesn't work, or these cells have a dysregulated BRCA1-53BP1 regulation).

What are the dissociation kinetics of DHX9:Pol II in relation to ongoing resection and later recombination?

This is an excellent study, with some important observations. It is somewhat superficial in places relying heavily on immunohistochemistry, I feel that a further exploration of the mechanisms, even if negative, is warranted.

Reviewer #3 (Remarks to the Author):

The authors identify a novel role for the RNA helicase DHX9 in homologous recombination repair. DHX9 and other RNA binding proteins have been recognized as important players in genome stability maintenance for some time, but the mechanisms have remained elusive. Through a series of experiments, the authors initially observed that DHX9 forms foci in response to damaging agents, which colocalize with gamma-H2AX, suggesting accumulation at the site of DNA double-strand breaks. Furthermore, depletion of H2AX lead to hypersensitivity to treatment with camptothecin, and inhibited formation of both RPA and RAD51 foci suggesting a role in early steps of HR repair. Genetic analysis indicated that H2AX acts in the same pathway as BRCA1, ATR, CtIP and MRE11. Co-immunoprecipitation experiments further showed that DHX9 preferentially interacts with BRCA1 in response to CPT treatment, and that this interaction was required for BRCA1 association with RNAP II. Together this data suggests a crucial role for DHX9 in homologous recombination repair both in recruitment of BRCA1 and in promoting end resection, a key early step in HR.

Overall the manuscript is interesting, clear, and identifies a novel mode of action for DHX9 in DSB repair. Nonetheless, there are several inconsistencies and points that require clarification and possibly new data to explain the authors' model. Changes to statistical analysis suggested below are especially important. Below I will highlight major issues, followed by a series of minor suggestions.

Major points:

#1 - The lack of effect of RNaseH1 treatment on DHX9 foci is somewhat surprising given the reported relationships of DHX9 to R-loops (the authors own work, and Cristini et al., 2018), and the roles of DNA:RNA hybrids in homologous recombination repair (PMIDs: 30245011; 32375052; 32555206; 30560944). Unfortunately the data with RNaseH1 is not shown and cannot be assessed. The authors should show the data and explain the discrepancy, if they feel there is one, in the context of the literature.

#2 - Line 95-99: The results here suggest that DHX9 foci formation could be a late event in HR due to the delay in formation of DHX9 foci compared to gamma-H2AX (at least 1.5 hours). However, the later results in the paper suggest that DHX9 is involved in some of the early processing steps such as RPA binding, and BRCA1 recruitment. The timeline of H2AX foci formation and other events in HR should be clarified to explain this contradiction. A time course of DHX9 localization in comparison to BRCA1, gH2AX and RPA should be shown after camptothecin treatment to support the claims on line 95.

This timing issue is doubly confusing in relation to the superficial analysis of ATM/ATR signaling in Figure 5. It would be nice to look at the effects of DHX9 on ATR autophosphorylation and on substrates (such as RPA, CtIP, BRCA1) to determine the steps at which the pathways overlap. Based on the gH2AX data, DHX9 is not completely required for ATM/ATR activation so some downstream step must be blocked? Conversely, the authors should test the effects of ATM and ATR inhibitors on DHX9 foci formation. Figuring out the order of steps will help to clarify the timing of DHX9 function in the pathway.

#3 - Line 103: The difference between Camptothecin induced damage and IR induced damage is central to the paper and revisited at Line 123, Line 177, and Line 306. While you explain how Camptothecin-induced damage is connected to transcription and replication, I think a similar brief discussion of IR induced damage and NHEJ here would be helpful. It needs to be very clear to the reader why this difference is important.

Specifically, the authors hint at the role of transcription as being the important difference in CPT treated cells. However, random IR-induced damage should hit some transcribed regions. Indeed, PolIII transcribes a significant proportion of the genome and therefore many of the IR breaks should be in transcribed regions. How does their model account for this? Are IR breaks fundamentally different? Additional discussion and possible experiments comparing agents like bleomycin or interstrand crosslinkers could help to resolve this.

#4 - Furthermore, Figure 5 showed induced interaction between DHX9 and BRCA1 under both CPT and IR treatment. However IR did not induce DHX9 localization to DSBs in Figure 1B, and si-DHX9 cells were hypersensitive to CPT but not IR. So is the DHX9-BRCA1 complex induced by IR not important? Overall, I am still not clear on the proposed role for the BRCA1-DHX9 complex. These kinds of discrepancies must be considered to fully account for the authors data.

#5 - How does DHX9 influence NHEJ? The authors showed that knockdown DHX9 significantly enhanced NHEJ repair pathway, suggesting an effect on pathway choice but then found that knockdown of 53BP1 did not suppress the HR defect in si-DHX9 cells.

#6 - Figure 3: The authors should indicate what %GFP is measuring on each graph since it changes between A and B. The difference between siControl and the positive controls in the NHEJ assay seem very small. Indeed, all of the effects in Figure 3 are small. The authors should show the replicate data points to help reviewers assess the reproducibility of the effects. More importantly, the authors used a one-way ANOVA to establish whether ANY significant differences exist between the datasets, but they do not report a post-hoc test to determine which differences are the significant ones. This must be corrected and the authors may find that not all of the differences are $p < 0.0001$ as reported. This is the case for EVERY figure in which one-way ANOVA is used, additional tests are required to put p-values on specific comparisons.

#7 - The 30 minute time point in 4B could reveal something interesting. If it were not for the four cells with values around 200 the amount of foci would be about the same between ctrl and siDHX9. Are these four cells outliers? Perhaps a larger sample size would clarify this. Regardless, if they were the same or similar then it suggests that rapid early accumulation of RPA foci may be unaffected, and only later accumulation of RPA is affected, perhaps on more extended resection tracks, or at sites that are delayed in breaking due to other processing. I feel like these observations could be investigated more deeply, especially given the confusion around the timing of DHX9 recruitment and its potential, along with BRCA1, to have impacts both early in HR repair through 53BP1 inhibition, and later in regulating resection.

Other Revisions:

- Line 199/Figure 5: Genetic analysis using RPA foci as a readout is used several times throughout the paper. I understand the premise of this analysis, but I think in all cases it was found that DHX9 acted in the same pathway as the other proteins tested. The authors should show a control of two proteins that induce RPA foci through different pathways to demonstrate that this assay is sensitive enough to detect additive changes in RPA foci.

- Despite the logic in the proposed naming convention for a new BRCA1-D complex (with DHX9), the authors should NOT try to take the BRCA1-D name when it already exists. This will only add confusion to the literature. I suggest the BRCA1-D2, BRCA1-Dh, or BRCA1-D9 complex instead.

- Paragraphs at Line 51 and Line 62: I think some of the information in these two paragraphs could be combined to make a more cohesive story. It feels like a list of facts, so it would be helpful to anchor it. Introduce your hypothesis (line 62) earlier and then explain how the known functions of DHX9 and current evidence lead you to pursue it as a potential DNA damage response protein.
- Line 189: I am unclear on what the statement "more than 50% of these RPA foci were also dependent on DHX9" is referring to.
- Statistical comparisons of the survival data in Figure 2 must be shown.
- Line 16: The sentence starting "Here..." seems to be missing a word. Possibly, "Here we..."?
- Line 135: "Whilst..." sentence is difficult to read.
- In Figure 2D the y-axis shows % cell growth, not surviving fraction as in 2B and 2C. Why? This is needlessly confusing.
- The authors state that DHX9 is essential in mouse. Does si-DHX9 alone (without CPT or olaparib or IR) affect cell viability? Could any such cytotoxicity influence the results? If DHX9 knockdown has no fitness defect, why not?
- The statistics in Figure 6 may be inappropriate as multiple comparisons are being done. An ANOVA with appropriate post hoc test is required instead of multiple t-tests. Given the small differences this may have an effect on the authors conclusions.
- siRNA knockdown efficiency of CTIP, MRE11, BRCA1, 53BP1, ATM, ATR and double knockdown efficiency western blots must be shown. Does knockdown any of these components affect cell viability, especially double KD conditions?
- Figure 6b- How did the authors calculate % colocalization between BRCA1 & DHX9 in untreated condition? Fig1A showed almost no DHX9 foci in untreated cells.
- RNaseA significantly decreased BRCA1 and DHX9 nuclear foci but did not affect the DHX9-BRCA1 complex (Fig7A IP experiment). So DHX9 and BRCA1 form a complex prior to localizing to sites of DNA damage?
- A western blot showing expression level of WT and helicase-dead mutant DHX9 plasmids should be included.
- The concentration and duration of DRB treatment should be indicated.
- In supplementary Figure 1G, siDHX9 cells after CPT treatment still generated a significant amount of ssDNA by BrdU staining. These effects should track with RPA foci formation but the magnitude of the effects do not really track very closely, why?
- In supplementary Figure 3B the authors claim that overexpression of WT DHX9 restored BRCA1 nuclear foci in si-DHX9 cells under CPT treatment. This could be rephrased since only 50% of the foci were restored.

We thank all the reviewers for their insightful comments that, we believe, have helped us to significantly improve our manuscript. We have done our best to provide all the extra data required and improvements to the text. Below we provide detailed responses to all the points raised.

Reviewer #1 (Remarks to the Author):

It has been reported recently that HR is somehow targeted to highly transcribed regions of the genome, while in the absence of transcription, HR is not targeted. In this report, the authors address the question of how HR is targeted to transcribing regions of the genome.

Experiments are conducted, and a critical role for the DHX9 gene is identified in the repair of DSBs by HR in a transcription and RNA dependent manner. The authors conduct a logical series of experiments that have been carefully devised, well controlled and are statistically robust, in terms of what is reported. The authors demonstrate that DHX9 defective cells fail to recruit RPA and RAD51 proteins to DNA damage sites and subsequently are unable to repair DSBs by HR. The experiments described are consistent with the interpretation that DHX9 plays a direct role in the repair of DSBs during HR by recruiting BRCA1 to RNA as a component of the RNA Polymerase II complex in the vicinity of DSB. Importantly, they provide good evidence that DHX9's role in the repair of DSBs is its ability to promote end-resection - an early step in the process of DSB repair by HR.

We thank this reviewer for their generous comments.

I have the following suggestions to improve the MS:

1. Given what is being reported about the mechanism, including the key elements that remain to be determined, it would be most helpful to have a current model of the proposed mechanism of DHX9 function in repair of DSBs by HR, as an aide to define future studies.

This is a helpful suggestion and we have now provided a model explaining how we think DHX9 contributes to HR in Supplementary Fig 5D

2. A clearer description of the significance and mechanisms of break induction by the various sources of DSB generating treatments used would be beneficial. Some of the significance of the different break-inducing treatments will not be so clear to the general reader.

We also agree with this suggestion. We have now made this clearer in the text (lines 86-94) and also provided a summary diagram to help the reader (Supplementary Fig 1)

3. There is a slight discrepancy regarding whether DHX9 plays a late or early role in the processing and repair of DSBs. The authors eventually conclude that it plays an early role in break resection. However, earlier in the Results section it is suggested that it might have a later role in break repair, based on its break recruitment timing. Alternative explanations are considered by the authors, but this discrepancy should be discussed in more detail to avoid possible confusion.

Thank you, we agree. We went back to our data to seek more clarity. What we see is that at early time points DHX9 does accumulate in foci, albeit they are very much fainter than those at later time points. This confirms that DHX9 is responding more rapidly to DNA damage than we previously thought and is now more consistent with timing of DNA end resection. We believe that the more intense foci that appear after longer camptothecin treatment are linked to the accumulation of stalled transcription complexes, which contain DHX9, at sites of prolonged DSB. We have altered the text accordingly (lines 108-119) and hope that this is clearer to the reader.

The manuscript is very well written, with only minor errors that should be eliminated during final editing of the report.

Thank you

Reviewer #2 (Remarks to the Author):

The manuscript makes a strong case for DHX9 involvement in the resection step of homologous recombination. The data illustrates it is recruited to sites of damage in an RNA-dependent fashion. DHX9 interacts both with BRCA1 and RNA Pol II, indeed, surprisingly its interaction with Pol II depends on the presence of BRCA1. Whether RNA polII : DHX9 :BRCA1 recruits as a unit, as suggested is difficult to gauge as only Camptothecin-treated cells have been examined for complex – is it pre-existing or induced?

A controversial and pressing question of how BRCA1 is recruited to DNA double strand breaks has been open for years; the data makes the case that RNA and DHX9 is central to BRCA1 recruitment. BRCA1 recruitment depends both on DHX9, RNA, and surprisingly, the unwinding activity of DHX9. It would be intriguing to understand further why the unwinding activity is required, since interaction between DHX9 and BRCA1 C terminus previously been described, and here RNA and its unwinding is further required.

A good deal of epistatic detail is given between BRCA1 and DHX9, and further ATR epistasis is consistent with a role on long range HR, consistent with BRCA1 regulation. However the fact that 53BP1 cannot rescue RPA foci or Olaparib resistance of DHX9 depleted cells, suggests that the role of DHX9 in HR is additional to BRCA1 recruitment. I'm not sure it's enough not to explore the facile explanations – poor Exo1/ DNA2:BLM recruitment, an examination of 53BP1-Rif1, are they as expected?

We thank this reviewer for their generous comments.

We believe that BRCA1 and DHX9 most likely exist as a preformed complex as RNA is not required to detect their interaction. We also know from our previous work that DHX9 is easier to detect, by both IP and fluorescence, under conditions where RNA Pol II is stalled. We hypothesize that this somehow stabilizes the complex. In the text we state that we do believe that DHX9 -BRCA1 exists as a preformed single unit that binds to RNA (line 287-289).

We hypothesize that the helicase domain of DHX9 is required for binding and unwinding of nascent RNA associated with RNA Pol II as we have reported previously. BRCA1 is recruited through its interaction with DHX9 and also contributes to the stability of this complex with RNA Pol II as shown by our IP (Fig 7b). Previous biochemical studies did not specifically address the contribution of the helicase domain or RNA in the recruitment of BRCA1 and DHX9 to RNA Pol II. We agree that more biochemical analysis of these critical interactions is required going forward.

The reviewer is right that DHX9 is required for more than suppression of 53BP1 mediated end joining, although this role in pathway choice remains important. We took the reviewers advice and explored this further. We now include evidence that RIF 1 is acting normally in that its recruitment is suppressed by DHX9 (as it is by BRCA1). Consequently knockdown of DHX9 leads to elevated recruitment of RIF1 to damaged chromatin (Fig 8A, lines 313-321). We also show that the recruitment of both CtIP and BLM are impaired in cells lacking DHX9 by IF and in cell extracts. Although not completely absent this is likely to be sufficient to decrease the efficiency of DNA

resection and explain why DHX9 is required for more than pathway choice alone (Fig 9 a-c, lines 331-339).

Lastly we demonstrate that DHX9 contributes with BRCA1 to the processing of Topol blocked DNA ends to enable replication restart and that this function is independent of 53BP1 status (Fig9d, lines 351-375). We do not claim that these findings completely or uniquely account for the role of DHX9 in HR but they do add additional strong support for our hypothesis that DHX9 plays a critical role in BRCA1 mediated DSB repair by HR

A positive control for Fig 8 a and b in which RPA foci and Olaparib resistance are restored in BRCA1 deficient cells by 53BP1 siRNA is needed (a facile explanation it that the siRNA doesn't work, or these cells have a dysregulated BRCA1-53BP1 regulation).

Thank you. We have provided the appropriate BRCA1/53BP1 control for Olaparib sensitivity (Fig 8c) and for resection (Fig 8b).

What are the dissociation kinetics of DHX9:Pol II in relation to ongoing resection and later recombination?

Given that the detection of DHX9/BRCA1 with RNA Pol II is affected by RNA Pol II stalling (as described above) and also that we believe that these proteins interact with RNA Pol II during normal transcription, we feel that attempting to measure DHX9:Pol II dissociation kinetics would not be straightforward for this study, nor would it strengthen the message.

This is an excellent study, with some important observations. It is somewhat superficial in places relying heavily on immunohistochemistry, I feel that a further exploration of the mechanisms, even if negative, is warranted.

Thank you.

Reviewer #3 (Remarks to the Author):

The authors identify a novel role for the RNA helicase DHX9 in homologous recombination repair. DHX9 and other RNA binding proteins have been recognized as important players in genome stability maintenance for some time, but the mechanisms have remained elusive. Through a series of experiments, the authors initially observed that DHX9 forms foci in response to damaging agents, which colocalize with gamma-H2AX, suggesting accumulation at the site of DNA double-strand breaks. Furthermore, depletion of H2AX lead to hypersensitivity to treatment with camptothecin, and inhibited formation of both RPA and RAD51 foci suggesting a role in early steps of HR repair. Genetic analysis indicated that H2AX acts in the same pathway as BRCA1, ATR, CtIP and MRE11. Co-immunoprecipitation experiments further showed that DHX9 preferentially interacts with BRCA1 in response to CPT treatment, and that this interaction was required for BRCA1 association with RNAP II. Together this data suggests a crucial role for DHX9 in homologous recombination repair both in recruitment of BRCA1 and in promoting end resection, a key early step in HR.

Overall the manuscript is interesting, clear, and identifies a novel mode of action for DHX9 in DSB repair. Nonetheless, there are several inconsistencies and points that require clarification and possibly new data to explain the authors' model. Changes to statistical analysis suggested below are especially important. Below I will highlight major issues, followed by a series of minor suggestions.

We thank this reviewer for their positive comments and have done our best to respond to their helpful suggestions

Major points:

#1 - The lack of effect of RNaseH1 treatment on DHX9 foci is somewhat surprising given the reported relationships of DHX9 to R-loops (the authors own work, and Cristini et al., 2018), and the roles of DNA:RNA hybrids in homologous recombination repair (PMIDs: 30245011; 32375052; 32555206; 30560944). Unfortunately the data with RNaseH1 is not shown and cannot be assessed. The authors should show the data and explain the discrepancy, if they feel there is one, in the context of the literature.

See below

#2 - Line 95-99: The results here suggest that DHX9 foci formation could be a late event in HR due to the delay in formation of DHX9 foci compared to gamma-H2AX (at least 1.5 hours). However, the later results in the paper suggest that DHX9 is involved in some of the early processing steps such as RPA binding, and BRCA1 recruitment. The timeline of H2AX foci formation and other events in HR should be clarified to explain this contradiction. A time course of DHX9 localization in comparison to BRCA1, gH2AX and RPA should be shown after camptothecin treatment to support the claims on line 95.

We address points #1 and #2 here:

Taking point 2 first. (Please also see response to reviewer 1.) We have now clarified that DHX9 is recruited earlier to DNA damage than we previously suggested but that this is manifest as very faint foci. Subsequently DHX9 accumulates to form bright foci. This now resolves the discrepancy between the apparent later appearance of DHX9 in foci compared to its proposed earlier role in resection. We have explained this more fully in the text (lines 108-119).

Point 1. This is a very good point. We did not specifically address the role of R-loops in HR in this manuscript given that it is somewhat controversial. However, since we treated the cells with RNaseH1 the reviewer is absolutely correct that we need to address the issue more clearly.

We find that at early time points the faint DHX9 foci are not sensitive to RNaseH1 treatment suggesting no link to R-loops. However, after prolonged treatment with Cpt the brighter DHX9 foci that form are diminished by RNaseH1 treatment, suggesting a contribution of DNA-RNA hybrid/R-loops. As described above we believe the intense DHX9 foci represent sites of stalled RNA polymerase II which is known to promote the formation of R-loops. It is also consistent with our previous work showing that DHX9 plays a critical role in R-loop formation. We now explain this more fully in the text (lines 121-134 and 409-418). We have not however, commented on whether these R-loops are required for HR mediated DSB repair because this too is controversial and in the current work we cannot distinguish whether they are simply a consequence of stalled RNA Pol or also required to perform HR.

The Reviewer drew attention the paper of Cristini et al which proposes that DHX9 promotes resolution of R-loops. However, we do not agree with Cristini on this matter and refer to our own work showing that DHX9 is required to generate R-loops (doi: 10.1038/s41467-018-06677-1). There are several reasons why we question the Cristini paper. Firstly, DHX9 functions with the wrong polarity (3'-5') to unwind canonical R-loops as depicted in most papers (i.e with a 5' RNA tail). Secondly, biochemical evidence shows that DHX9 preferentially binds ssRNA and unwinds RNA

secondary structures, which is also consistent with the 2018 paper by Aktas et al doi: 10.1038/nature21715. Finally, data in the Cristini paper itself shows that knockdown of DHX9 reduces global DNA-RNA hybrid.

This timing issue is doubly confusing in relation to the superficial analysis of ATM/ATR signaling in Figure 5. It would be nice to look at the effects of DHX9 on ATR autophosphorylation and on substrates (such as RPA, CtIP, BRCA1) to determine the steps at which the pathways overlap. Based on the gH2AX data, DHX9 is not completely required for ATM/ATR activation so some downstream step must be blocked? Conversely, the authors should test the effects of ATM and ATR inhibitors on DHX9 foci formation. Figuring out the order of steps will help to clarify the timing of DHX9 function in the pathway.

We thank the reviewer for this very good suggestion. We addressed the issue of the timing of DHX9 function above. However, we agree that the relationship with ATR and ATM merited further analysis. We have now included a figure to show how ATM and ATR inhibitors impact on DHX9 foci (Fig 5c). We find that DHX9 foci are dependent on ATR signalling and partially on ATM. This supports our previous assertion that these proteins function in the same pathway for resection (Fig 5d). Our new data shows that DHX9 also promotes ATR auto-phosphorylation (and Chk1 phosphorylation) suggesting is a potentially inter-dependent relationship between DHX9 and ATR (Fig 5e). We have not stressed this as it is the subject for ongoing study to determine the mechanism through which this operates.

#3 - Line 103: The difference between Camptothecin induced damage and IR induced damage is central to the paper and revisited at Line 123, Line 177, and Line 306. While you explain how Camptothecin-induced damage is connected to transcription and replication, I think a similar brief discussion of IR induced damage and NHEJ here would be helpful. It needs to be very clear to the reader why this difference is important.

Specifically, the authors hint at the role of transcription as being the important difference in CPT treated cells. However, random IR-induced damage should hit some transcribed regions. Indeed, PolIII transcribes a significant proportion of the genome and therefore many of the IR breaks should be in transcribed regions. How does their model account for this? Are IR breaks fundamentally different? Additional discussion and possible experiments comparing agents like bleomycin or interstrand crosslinkers could help to resolve this.

We agree with the reviewer and have now expanded our explanation of IR and Cpt induced damage, and their relationship to transcription in the results section (Supplementary Fig1, lines 86-94, 116-119) and in the discussion (lines 400-407). We think that introducing more agents might actually complicate things at this point. We intend further studies try to specifically link DHX9 to transcription associated HR, including looking at the proportion of IR induced DNA breaks which might require repair by this pathway. However, at this point our goal is to establish that DHX9 plays a critical role in DSB repair by HR.

#4 - Furthermore, Figure 5 showed induced interaction between DHX9 and BRCA1 under both CPT and IR treatment. However IR did not induce DHX9 localization to DSBs in Figure 1B, and si-DHX9 cells were hypersensitive to CPT but not IR. So is the DHX9-BRCA1 complex induced by IR not important? Overall, I am still not clear on the proposed role for the BRCA1-DHX9 complex. These kinds of discrepancies must be considered to fully account for the authors data.

Thank you for this comment. We acknowledge we did not explain this sufficiently clearly. It is known that the majority of Cpt induced lesions (DSB) are linked with transcription because of the way they

are generated (inhibition of Topo1). Consequently, DHX9 co-localizes significantly with DSB generated through this mechanism. The reviewer is correct that DSB caused by IR may also induce lesions in transcribed regions but since the generation of these DSB are not dependent on transcription they are different. For instance they are likely to be two ended rather than one ended breaks and therefore amenable to NHEJ. We speculate that the DHX9 foci in IR treated cells are also linked to stalled or slowed transcription complexes (since they too are dependent on RNA) but perhaps caused by IR induced lesions other than dsb, such as ssbreaks or adducts. Consequently both Cpt and IR enhance detection of DHX9-BRCA1 as both stall transcription but those caused by Cpt are more closely linked to DSB and therefore co-localize with γ H2AX. We have tried to make this clearer in the text.

#5 - How does DHX9 influence NHEJ? The authors showed that knockdown DHX9 significantly enhanced NHEJ repair pathway, suggesting an effect on pathway choice but then found that knockdown of 53BP1 did not suppress the HR defect in si-DHX9 cells.

We have shown that DHX9 plays a role in pathway choice by including new data demonstrating that like BRCA1 it suppresses recruitment of Rif1 to damage (Fig 8a). Since knockdown 53BP1 does not suppress the HR defect in DHX9 cells, as it does for BRCA1, we hypothesized that DHX9 plays an addition role to help to promote DNA resection. Although we cannot claim to have a fully established mechanism for this, we now present evidence that DHX9 is required for the efficient recruitment of the DNA resection proteins CtIP and Blm to chromatin in response to DNA damage (Fig 9a-c). Lastly we have shown that DHX9 also plays a role in the processing of Cpt-induced DNA breaks to support replication restart, a function that is independent of 53BP1 and pathway choice (Fig 9d).

#6 - Figure 3: The authors should indicate what %GFP is measuring on each graph since it changes between A and B. The difference between siControl and the positive controls in the NHEJ assay seem very small. Indeed, all of the effects in Figure 3 are small. The authors should show the replicate data points to help reviewers assess the reproducibility of the effects. More importantly, the authors used a one-way ANOVA to establish whether ANY significant differences exist between the datasets, but they do not report a post-hoc test to determine which differences are the significant ones. This must be corrected and the authors may find that not all of the differences are $p < 0.0001$ as reported. This is the case for EVERY figure in which one-way ANOVA is used, additional tests are required to put p-values on specific comparisons.

Thank you we have now included a diagram to explain the different assays and the different mechanisms for the generation of GFP (Fig 3a-b). We have also included a Tukey's post-hoc test to establish significance between data sets, for this and all ANOVA tests.

#7 - The 30 minute time point in 4B could reveal something interesting. If it were not for the four cells with values around 200 the amount of foci would be about the same between ctrl and siDHX9. Are these four cells outliers? Perhaps a larger sample size would clarify this. Regardless, if they were the same or similar then it suggests that rapid early accumulation of RPA foci may be unaffected, and only later accumulation of RPA is affected, perhaps on more extended resection tracks, or at sites that are delayed in breaking due to other processing. I feel like these observations could be investigated more deeply, especially given the confusion around the timing of DHX9 recruitment and its potential, along with BRCA1, to have impacts both early in HR repair through 53BP1 inhibition, and later in regulating resection.

We take the reviewers point here. The Mann-Whitney test that we employed is common for this type of data. It is a rank test that speaks to the independence / difference between samples. It does not specifically address differences between means. To address this we have now also performed an ANOVA with Tukey's post-hoc test which alleviates the discrepancy noted by the reviewer. This confirms that the differences in resection observed between cells that are proficient and deficient in DHX9 are first apparent somewhere between 30 mins and an hour. We cannot say anything about whether these later resection tracks are more extended because we did not measure intensity of individual foci and I am not sure that this method be the best way to measure extent of resection. This is likely something we will pick up again in onward studies.

Other Revisions:

- Line 199/Figure 5: Genetic analysis using RPA foci as a readout is used several times throughout the paper. I understand the premise of this analysis, but I think in all cases it was found that DHX9 acted in the same pathway as the other proteins tested. The authors should show a control of two proteins that induce RPA foci through different pathways to demonstrate that this assay is sensitive enough to detect additive changes in RPA foci.

We understand the point the reviewer is making here. However, we were unable to think of defects in any two proteins that would give an additive effect for resection. In other words, for HR there is only one pathway for resection and therefore epistasis is likely for almost any combination of knockdowns in resection factors. We know that the assay is sensitive enough because in our previous work where RPA accumulated as a consequence of stalled replication complexes (through ssDNA spooling) and also fork collapse to produce dsb, we were able to show that knockdown of BRCA1/CtIP mediated resection was responsible of only 1/3 of RPA foci. In the current work, however, it is clear that all RPA foci are dependent on resection.

- Despite the logic in the proposed naming convention for a new BRCA1-D complex (with DHX9), the authors should NOT try to take the BRCA1-D name when it already exists. This will only add confusion to the literature. I suggest the BRCA1-D2, BRCA1-Dh, or BRCA1-D9 complex instead.

On the advice of the reviewer we have re-designated the complex as BRCA1-DX. It's a shame that the term BRCA1-D nomenclature was incorrectly assigned to another complex that did not specifically involve a protein interaction mediated by the BRCT domain of BRCA1 that is involved in resection, as is the case for BRCA1-A -B and- C. We thought that since this designation was only used once in a review we might use the BRCA1-D nomenclature correctly. But we agree with the reviewer that this might cause unnecessary confusion.

- Paragraphs at Line 51 and Line 62: I think some of the information in these two paragraphs could be combined to make a more cohesive story. It feels like a list of facts, so it would be helpful to anchor it. Introduce your hypothesis (line 62) earlier and then explain how the known functions of DHX9 and current evidence lead you to pursue it as a potential DNA damage response protein.

We have tried to reorganize this section so that it flows better

- Line 189: I am unclear on what the statement "more than 50% of these RPA foci were also dependent on DHX9" is referring to.

Sorry for not being clear, we have deleted this statement and replaced with a less confusing statement based on our data for both RPA and BrdU (lines 208-209).

- Statistical comparisons of the survival data in Figure 2 must be shown.

We have now done this

-Line 16: The sentence starting “Here...” seems to be missing a word. Possibly, “Here we...”?

Thank you, we have now corrected this.

-Line 135: “Whilst...” sentence is difficult to read.

Thank you. We have now rewritten the sentence and hope that it is clearer

-In Figure 2D the y-axis shows % cell growth, not surviving fraction as in 2B and 2C. Why? This is needlessly confusing.

The reason for this is that figures in 2b and 2b are clonogenic survival assays. Fig 2d shows the data for a cell viability/proliferation assay. We initially performed all assays using the cell viability method, which is very reliable and been used by many groups. Since we did not see any additional sensitivity with IR we did not perform an additional clonogenic assay.

-The authors state that DHX9 is essential in mouse. Does si-DHX9 alone (without CPT or olaparib or IR) affect cell viability? Could any such cytotoxicity influence the results? If DHX9 knockdown has no fitness defect, why not?

In our previous study we showed that cell proliferation and viability was fine in cells knocked down for DHX9. We assume that a residual amount of helicase is sufficient to keep cells growing while enabling us to look at its contribution to various cell functions. In fact in our previous work, knockdown of DHX9 restored viability caused by other defects (doi: 10.1038/s41467-018-06677-1).

- The statistics in Figure 6 may be inappropriate as multiple comparisons are being done. An ANOVA with appropriate post hoc test is required instead of multiple t-tests. Given the small differences this may have an effect on the authors conclusions.

We agree and have re-evaluated with ANOVA plus Tukey's post hoc test

- siRNA knockdown efficiency of CTIP, MRE11, BRCA1, 53BP1, ATM, ATR and double knockdown efficiency western blots must be shown. Does knockdown any of these components affect cell viability, especially double KD conditions?

Yes we agree and this information is now provided in Supplementary Fig Westerns. We have not detected any decrease in cell viability in any of our knockdowns that impacts on the observations of our study. In some cases (CtIP, Mre11) this is likely to result from residual low level protein expression. SiRNA knockdown of all these proteins have also been reported previously by others.

- Figure 6b- How did the authors calculate % colocalization between BRCA1 & DHX9 in untreated condition? Fig1A showed almost no DHX9 foci in untreated cells.

Colocalization was measured by Pearson correlation coefficient which considers co-occurrence (simple spatial overlap) and correlation (the co-distribution) of signal within and between structures i.e. foci. The programme takes regions within a single cell and measures these two criteria whether or not there is a high signal intensity or there is simply background signal. In cells with little or no signal (as in untreated sample) there could still be colocalization (if co-occurrence and correlation were established) but in our case there is not as the PCC for untreated cells is 0.1 – so little or no meaningful colocalization.

- RNaseA significantly decreased BRCA1 and DHX9 nuclear foci but did not affect the DHX9-BRCA1 complex (Fig7A IP experiment). So DHX9 and BRCA1 form a complex prior to localizing to sites of DNA damage?

The reviewer is correct in their conclusion. We did indicate this in the original manuscript but have now made this clearer. We believe that BRCA1-DHX9 forms independently of RNA but binds nascent RNA as a preformed unit. The detection of this complex is enhanced when RNA pol II is stalled, for example at DNA damage.

- A western blot showing expression level of WT and helicase-dead mutant DHX9 plasmids should be included.

Thank you. We agree and have provided this in Supplementary Fig Westerns

- The concentration and duration of DRB treatment should be indicated.

Sorry we should have included this and have corrected the omission

- In supplementary Figure 1G, siDHX9 cells after CPT treatment still generated a significant amount of ssDNA by BrdU staining. These effects should track with RPA foci formation but the magnitude of the effects do not really track very closely, why?

We believe that the BrdU data are consistent with the RPA data for resection. In the absence of DNA damage there only a baseline level of BrdU foci, which increase in cells treated with camptothecin. Cells defective in DHX9 are impaired in the formation of BrdU foci even after DNA damage. This directly mirrors our RPA data. While we appreciate for BrdU foci that there is an increased spread amongst the population of DHX9 defective cells treated with Cpt, this might be explained by a minority of cells that are not fully knocked down for DHX9 - most of the cells still cluster at the baseline level. Finally ANOVA with post-hoc test (as recommended by the reviewer) confirms that there is no significant difference in BrdU staining between DHX9 cells with or without DNA damage.

- In supplementary Figure 3B the authors claim that overexpression of WT DHX9 restored BRCA1 nuclear foci in si-DHX9 cells under CPT treatment. This could be rephrased since only 50% of the foci were restored.

Yes we agree and have rephrased this. Thank you

REVIEWERS' COMMENTS

Reviewer #1 (Remarks to the Author):

The authors have satisfactorily addressed all my initial comments in this much improved revision of the manuscript. I have no further comments or concerns regarding publication of the report.

Reviewer #2 (Remarks to the Author):

Many congratulations - a fascinating report.

Reviewer #3 (Remarks to the Author):

Thanks for addressing all of my comments thoughtfully. A few outstanding issues.

The supplementary material is out of order or missing information. Perhaps this was a PDF conversion issue but the legend for Figure S2 appears on a page with Figure S4 in the corner. Also the supplementary gels have no figure number (presumably Figure S6), it just says supplementary gels. Anyway, I encourage the authors to ensure all of the information is accurately labelled and ordered to enhance readability.

I also disagree with the DHX9 polarity argument to refute its potential as an R-loop unwinding enzyme (3'-5'). The cartoon drawings may not always reflect reality. Does DHX9 require a free end to work? It seems unlikely that the 5' end of the RNA will be annealed to the genome. There would just be junctions between dsDNA and the DNA:RNA hybrid at the 3' or 5' ends. In many cases the RNA polymerase will be further down the gene and I can't see why DHX9 couldn't recognize the 3'-hybrid dsDNA junction and act to unwind the R-loop. Anyway, fortunately this point was not stressed in the text so no revision is really required. I just encourage the authors to continue to think about this.

The explanation for a lack of fitness defect in siDHX9 cells compared to its essentiality in mice is also unconvincing. The argument seems to be that the knockdown is insufficient and some small amount of residual DHX9 is enough to fulfill some essential function. What is this essential function? Does it not relate to the roles of DHX9 in genome stability described here? Perhaps its more likely that some key developmental steps require high DHX9 activity but that this is dispensable in mature somatic cells. Or that DHX9 is dispensable only in the context of the transformed cancer cell lines here. Anyway, I just felt the authors own explanation of incomplete siRNA penetrance undermined their own work when other explanations were available. Again, these are issues with the rebuttal not the manuscript so should not have bearing on potential acceptance.

We thank all the reviewers for their efforts to improve our manuscript for publication in Nature Comms. We appreciate their constructive criticisms and helpful suggestions. We respond to final specific comments below.

Reviewer #1 (Remarks to the Author):

The authors have satisfactorily addressed all my initial comments in this much improved revision of the manuscript. I have no further comments or concerns regarding publication of the report.

Thank you very much

Reviewer #2 (Remarks to the Author):

Many congratulations - a fascinating report.

Thank you very much

Reviewer #3 (Remarks to the Author):

Thanks for addressing all of my comments thoughtfully. A few outstanding issues.

Thank you also for your helpful suggestions

The supplementary material is out of order or missing information. Perhaps this was a PDF conversion issue but the legend for Figure S2 appears on a page with Figure S4 in the corner. Also the supplementary gels have no figure number (presumably Figure S6), it just says supplementary gels. Anyway, I encourage the authors to ensure all of the information is accurately labelled and ordered to enhance readability.

Yes, we apologize, we have now corrected this

I also disagree with the DHX9 polarity argument to refute its potential as an R-loop unwinding enzyme (3'-5'). The cartoon drawings may not always reflect reality. Does DHX9 require a free end to work? It seems unlikely that the 5' end of the RNA will be annealed to the genome. There would just be junctions between dsDNA and the DNA:RNA hybrid at the 3' or 5' ends. In many cases the RNA polymerase will be further down the gene and I can't see why DHX9 couldn't recognize the 3'-hybrid dsDNA junction and act to unwind the R-loop. Anyway, fortunately this point was not stressed in the text so no revision is really required. I just encourage the authors to continue to think about this.

This is an important issue and is yet to be fully resolved in the field. For our part, we have tried to be as open minded as possible. However, weighing up current evidence we believe that it is unlikely that DHX9 resolves R loops by unwinding RNA-DNA hybrid (although we agree it is not impossible). Our current reasoning is as follows.

1. Biological evidence- in our previous publication (doi:10.1038/s41467-018-06677-1) we presented several pieces of cell biological and genetic evidence (immunofluorescence imaging, genetic suppression, ChIP and DRIP-PCR) that R-loops were diminished in the absence of DHX9 suggesting that they contribute to the generation and not the resolution of R-loops. Our current submitted work also shows DHX9 accumulates with increased DNA-RNA hybrid at dsb, consistent with generation and not the resolution of R-loops by DHX9. A different view was taken in the paper by Cristini et al (doi:10.1016/j.celrep.2018.04.025), who proposed that DHX9 resolved R-loops induced by camptothecin treatment. Even so, this

report also included data showing that R-loops (generally) are decreased in cells knocked down for DHX9 (Supp Figures S3, S5) indicating that DHX9 might also promote global R-loop formation under the conditions used in their study. Nevertheless, based on these two reports alone it is difficult to identify the source of the discrepancies.

2. Biochemical evidence- When considering the biochemical activities of DHX9, it is more difficult to see how it can unwind R-loops. We re-evaluated the data in the paper that first identified DHX9 as a potential R-loop resolving protein (doi: 10.1093/nar/gkq240). Interestingly, while DHX9 clearly unwinds RNA-DNA hybrids in vitro, it does so with a 3'-5' polarity, which presents a problem when considering unwinding of physiological R-loops.

To unwind/resolve an R-loop, DHX9 must load on and translocate along either ssRNA or ssDNA. The strongest evidence from in vitro biochemistry, and supported by the recent demonstration of a role for R-loops in resolving structures formed by inverted Alu repeats (doi: 10.1038/nature21715), is that DHX9 acts primarily on RNA substrates.

An R loop is thought to form when nascent RNA emerges from RNA Pol II under conditions where it can pair with its complementary ssDNA. It is of note that this nascent RNA is ssRNA with a 5' end. In the original paper (doi: 10.1093/nar/gkq240) DHX9 was unable to unwind such 'physiological type R-loops, but only R-loops with a 3' ssRNA end. Binding to a 5' ssRNA would result in translocation away from the region of DNA-RNA hybrid that required unwinding (scenario 1 below). Hence simple unwinding of a physiological R-loop by translocation of DHX9 along the free end of the nascent strand is not possible.

An alternative is that the nascent 5' RNA end is fully paired with DNA and DHX9 translocates 3'-5' along DNA to unwind the hybrid region (scenario 2). However, binding and translocation along even a short region of DNA has been shown to greatly impair DHX9 unwinding activity (DOI: 10.1021/acs.biochem.8b01025). It is also inconsistent with the in vivo activity of DHX9 as an RNA structure resolving enzyme. Hence we believe this mechanism for resolving R-loops is also very unlikely.

A third scenario is that backtracking of RNA Pol II results in the generation of an R-loop with a 3' ssRNA tail. DHX9 might unwind this structure but its complete resolution would require displacement or dissociation of the transcription complex from DNA and there is currently little evidence to support this. In fact persistence of RNA Pol II at R-loops potentially contributes to the pathology of R-loops. Again we feel it very unlikely that R-loops are resolved by this mechanism.

Finally, it is theoretically possible that DHX9 might binds to ssRNA 3' that exists between the advanced RNA Pol II and the region of DNA-RNA hybrid (scenario 4). If this extended region of ssRNA exists, DHX9 might bind but is very unlikely to unwind the hybrid region due to the lack of a free RNA end (since the 3' RNA end remains attached to RNA Pol II). In the absence of such an end, issues of torsion and rotation mean it is unlikely that DHX9 can unwind of the paired region. A similar requirement for a free ssRNA end has been shown for the generation of R-loops.

Taking all this into consideration; while it is not impossible for DHX9 to unwind a physiological R-loop, biochemical data is currently against this. If also we consider that at least one report (from our lab) finds that DHX9 defects are associated with decreased rather than increased global R-loops, it is hard to make a strong case for DHX9 as an R-loop resolving protein. On the other hand much of the data correlates well with our model that DHX9 promotes R-loop formation by unwinding RNA secondary structures (doi: 10.1093/nar/gkq240). Nevertheless we retain an open mind and as more evidence comes to light it is possible that our view will alter and we might explain the discrepancies between the Cristini and Chakraborty papers. I hope this clarifies our thinking for the reviewer.

The explanation for a lack of fitness defect in siDHX9 cells compared to its essentiality in mice is also unconvincing. The argument seems to be that the knockdown is insufficient and some small amount of residual DHX9 is enough to fulfill some essential function. What is this essential function? Does it not relate to the roles of DHX9 in genome stability described here? Perhaps its more likely that some key developmental steps require high DHX9 activity but that this is dispensable in mature somatic cells. Or that DHX9 is dispensable only in the context of the transformed cancer cell lines here. Anyway, I just felt the authors own explanation of incomplete siRNA penetrance undermined their own work when other explanations were available. Again, these are issues with the rebuttal not the manuscript so should not have bearing on potential acceptance.

We respectfully disagree with the reviewer here. Many essential proteins have been studied using incomplete knockdown by siRNA. In fact it can be quite difficult to completely nullify the activity of enzymes even when little or no protein can be detected (even with other methods such as auxin induced degradation). DHX9 knockout is lethal in all tissue culture cell lines tested to date (doi: 10.1016/j.cell.2015.11.015). Subsequent, to the publication of our pre-print, the Durocher lab published an unbiased Crispr knockout screen, in which DHX9 was found in the group of proteins conferring an HR defective phenotype (doi:10.1016/j.cell.2020.05.040). We suspect that lethality arises through a combination of impaired transcription and HR failure leading to increased replication stress. Indeed, we have found that depletion of various RNA binding proteins, with incomplete penetrance, reveals DNA repair defects, suggesting that impaired but not ablated RNA processing is linked to the maintenance of genomic integrity . However, we agree with the reviewer that this means one needs to be careful with interpretation.

We hope this explains our thinking on these two issues. As indicated by the reviewer, these issues are not critical for the current manuscript but are interesting and very important going forward.